# Chromatin organization drives the search mechanism of nuclear factors

Matteo Mazzocca [1,6], Alessia Loffreda [2,6], Emanuele Colombo[1], Tom Fillot[1,2], Daniela Gnani[1], Paola Falletta[1], Emanuele Monteleone [1], Serena Capozi[3], Edouard Bertrand [3], Gaelle Legube [4], Zeno Lavagnino[2,5], Carlo Tacchetti[1,2] & Davide Mazza [1,2] ✉

Nuclear factors rapidly scan the genome for their targets, but the role of nuclear organization in such search is uncharted. Here we analyzed how multiple factors explore chromatin, combining live-cell single-molecule tracking with multifocal structured illumination of DNA density. We find that factors displaying higher bound fractions sample DNA-dense regions more exhaustively. Focusing on the tumor-suppressor p53, we demonstrate that it searches for targets by alternating between rapid diffusion in the inter-chromatin compartment and compact sampling of chromatin dense regions. Efficient targeting requires balanced interactions with chromatin: fusing p53 with an exogenous intrinsically disordered region potentiates p53-mediated target gene activation at low concentrations, but leads to condensates at higher levels, derailing its search and downregulating transcription. Our findings highlight the role of disordered regions on factors search and show-case a powerful method to generate traffic maps of the eukaryotic nucleus to dissect how its organization guides nuclear factors action.

Advances in microscopy and biochemistry over the last two decades are shedding light on the organization principles of the eukaryotic cell nucleus. DNA/DNA, DNA/RNA, and DNA/protein interactions render the nucleus a highly compartmentalized organelle at multiple scales, with different compartments being specialized in specific tasks[1–3]. At the nanometer scale, DNA is wrapped into chromatin by nucleosomes, and the chromatin fiber is further organized into a hierarchy of higher-order structures giving rise to topologically associated domains and chromatin compartments, that cluster together genes that are similarly regulated[4]. At the sub-micron scale, the compaction of the chromatin fibers results in chromatin-dense (CD) domains proposed to contain inactive genes, interspersed by a branched network of chromatin-depleted channels, known as the interchromatin compartment (IC)[5]. Recent computational and experimental work highlights that both active transcription and large macromolecular complexes

involved in maintaining nuclear architecture are preferentially located at the CD/IC interface[6,7]. These observations have led to the hypothesis that the IC might serve as a route for targeting diffusible macro-molecules such as transcription factors (TFs) to their site of action (e.g., their binding sites on enhancer/promoters), potentially accelerating their target search mechanism[5].

A rapid search mechanism is particularly relevant for inducible TFs (iTFs) facing the apparently gargantuan task of identifying their target sites among billions of potential decoy sites[8,9]. Remarkably, many iTFs are capable of activating transcription of their target genes in minutes following the activating stimulus[10], a timescale that seems incompatible with the classical diffusion-limited search mechanism[11]. This target search problem has been addressed extensively in bacteria, where an efficient search is achieved by facilitated diffusion, a mechanism combining 1D sliding on DNA

[1]Università Vita-Salute San Raffaele, Via Olgettina 58, 20132 Milan, Italy. [2]IRCCS Ospedale San Raffaele, Experimental Imaging Center, Via Olgettina 58, 20132 Milan, Italy. [3]Institut de Génétique Moléculaire de Montpellier, CNRS, Montpellier 34293, France. [4]MCD, Centre de Biologie Intégrative (CBI), CNRS, Université de Toulouse, UT3, Toulouse, France. [5]Present address: IFOM ETS– The AIRC Institute of Molecular Oncology—Via Adamello 16, 20139 Milan, Italy. [6]These authors contributed equally: Matteo Mazzocca, Alessia Loffreda. ✉e-mail: mazza.davide@hsr.it

with 3D diffusion[12–14]. Whether TFs exploit a similar 1D + 3D search mechanism in the eukaryotic cell nucleus is unclear, as the large excess of non-specific binding sites, the heterogeneity in chromatin density, and the presence of nucleosomes might limit the efficacy of facilitated diffusion.

In recent years, single-molecule tracking (SMT)[15,16] has high-lighted some mechanisms used by nuclear proteins to search for their target sites in eukaryotes[17,18]. Live-cell SMT of proteins involved in the maintenance of chromatin organization (such as CTCF)[19] or its epigenetic state (such as the Polycomb protein Cbx2)[20,21] display an accelerated search mechanism named guided exploration[8,19]. Here, transient multivalent interactions mediated by the protein intrinsically disordered regions (IDRs) trap the factor in nuclear condensates or clusters that are explored in a compact (exhaustive) manner.

The role of condensates in transcription remains however fervently debated[22]. While some data point to condensates as a way to enhance the on-rate of TFs to their binding sites and consequently amplify transcription[23,24], recent findings highlight that condensates could also be detrimental to transcription[25]. On the other hand, a recent report on artificial TFs suggests that the IDRs per se, rather than condensates formation, could have a positive role in TF-binding and transcriptional activation[26].

In this context, whether guided exploration characterizes the dynamics of diverse nuclear proteins/TFs—especially of those not participating in the assembly of visible condensates—and how the nuclear organization affects the TF search remain fundamental, yet unaddressed, questions.

To tackle these questions, here we combine SMT of nuclear factors (NFs) with super-resolved structured illumination microscopy[27] (multifocal SIM, mSIM) of a reference channel in an integrated optical microscope to track single molecules within their chromatin environment with 150 nm resolution.

We exploit the developed SMT/mSIM microscope to characterize the role of chromatin in shaping the search mechanism of different nuclear proteins, showing that different factors display distinct accessibility to DNA-dense regions that strongly correlate with their propensity to interact with immobile scaffolds, such as chromatin itself. We next characterize the search mechanism of the tumor-suppressor p53, a key TF involved in the transcriptional response to DNA damage. Although p53 has been shown to be capable of facilitated diffusion in test tube experiments[28,29], its search mechanism remains unexplored in living nuclei. Our analysis shows that the p53 search is defined by rapid diffusion in the IC and slowed down anisotropic diffusion in CDs, where chromatin-bound p53 molecules are enriched. At the IC/CD boundary, p53 diffusion is slowed down and rendered more compact by interactions mediated by p53 IDRs.

To characterize the role of its IDRs in p53's target search mechanism we perform SMT on p53 mutants, either by deleting them, or by fusing an exogenous IDR–derived from the FUS protein -, in order to increase its multivalent interactions. IDR deletion mutants display reduced compact exploration and a lower capability to access high-DNA density regions, leading to lower expression of p53 target genes. On the other hand, the exogenous IDR from FUS can lead p53 to enhance the expression of its target genes. However, when expressed at concentrations sufficient to form condensates, FUS-p53 displays a derailed target search resulting in a drop in transcriptional activation. Our results suggest that condensates of transcriptional activators are not exclusively activating, but can also shut down transcription. We predict that the application of SMT/mSIM to multiple TFs and chromatin/organelles binding proteins will enable us to dissect how the crowded nuclear environment can guide the molecular players involved in transcription, DNA replication, and DNA repair to their targets.

## Results

### Different NFs have distinct accessibility to dense chromatin

To address how the search mechanism of nuclear factors (NFs) relates to chromatin organization, we designed a microscope capable of combining Highly Inclined Laminated Optical Sheet (HILO) imaging[30] of TF dynamics by SMT with super-resolution imaging of a reference channel, by multifocal structured illumination microscopy (mSIM, Supplementary Fig. 1a). As detailed in the Methods, mSIM imaging is based on scanning a pattern of illumination spots on the sample, created by a digital micromirror device (DMD). The image is then reconstructed via virtual pinholing (resulting in confocal-grade optical sectioning) and pixel reassignment to increase the lateral resolution. We confirmed that mSIM increases the lateral resolution close to the theoretical 1.4× maximum by imaging microtubules in fixed cells (Supplementary Fig. 1b). Additionally, we observed improved optical sectioning by imaging DNA stained with Hoechst 33342 in living osteosarcoma cells (Fig. 1a). Further, we evaluated that chromatic aberrations between the SMT and the mSIM channels were on average below the resolution limit (Supplementary Fig. 1c). Finally, we moved on combining mSIM of DNA density with single-molecule imaging of Halo-tagged proteins diffusing in the nucleus of cells (Supplementary Movie 1).

To this end, we recorded the positions of the detected single molecules for each of the tested proteins and measured the Hoechst intensity at each of these positions, classified in quartiles (Fig. 1b, c). We focused our analysis on one "inert" tracer (unconjugated HaloTag), two TFs (p53 and p65, a subunit of NF-kB), an architectural protein (CTCF), and a chromatin core component (Histone H2B). For ectopically expressed p65 and Histone H2B we verified that the expression levels of the exogenous protein did not exceed 1.5-fold of the endogenous ones (Supplementary Fig. 2a). The analysis of the SMT/mSIM data revealed that different factors map against DNA density with different likelihoods. Unconjugated HaloTag was found to be enriched in regions at lower DNA density, while histone H2B was preferentially found in regions with higher DNA density (Fig. 1c). The TFs and CTCF showed an intermediate profile, with CTCF displaying more enrichment in DNA-dense regions than p53 and p65 (Fig. 1d).

We next checked if these differences in inaccessibility to DNA-dense nuclear regions could be explained by simple size-exclusion effects that would predict larger factors to be more excluded from CD regions. However, we observed no correlation between protein molecular weight and enrichment in higher DNA classes (Fig. 1e)—even when factors oligomerization was taken into account (Supplementary Fig. 2b)—suggesting that additional mechanisms might be at play. Indeed, the enrichment of NFs in DNA-dense regions was found to correlate to their ability to bind chromatin, as measured by the fraction of bound molecules[31] (Fig. 1f). These results suggest that the different localization of the tested NFs might be driven by interactions with scaffolds, such as chromatin itself.

Next, we tested if the diffusion of each NF is controlled by the DNA density of the visited nuclear regions. To this end, we quantified the instantaneous diffusion coefficient $D_{inst}$ associated to each single-molecule displacement and evaluated the probability to observe a certain $D_{inst}$ in each DNA density class (Fig. 1g).

This analysis revealed classes of nuclear factors with strikingly different dynamic behaviors. H2B and p65 are enriched in DNA-dense and DNA-poor regions respectively, regardless of how fast they diffuse. In contrast, for both p53 and CTCF, slow-diffusing molecules are enriched in DNA-dense regions, while fast-diffusing molecules are enriched in DNA-poor regions. (Fig. 1g).

In sum, different NFs sense DNA density differently, with preferential localization at DNA-dense regions being apparently driven by their propensity to interact with nuclear scaffolds, as estimated by the single-molecule bound fraction. Further, some factors such as CTCF and p53 are slowed down at regions with higher DNA density, while

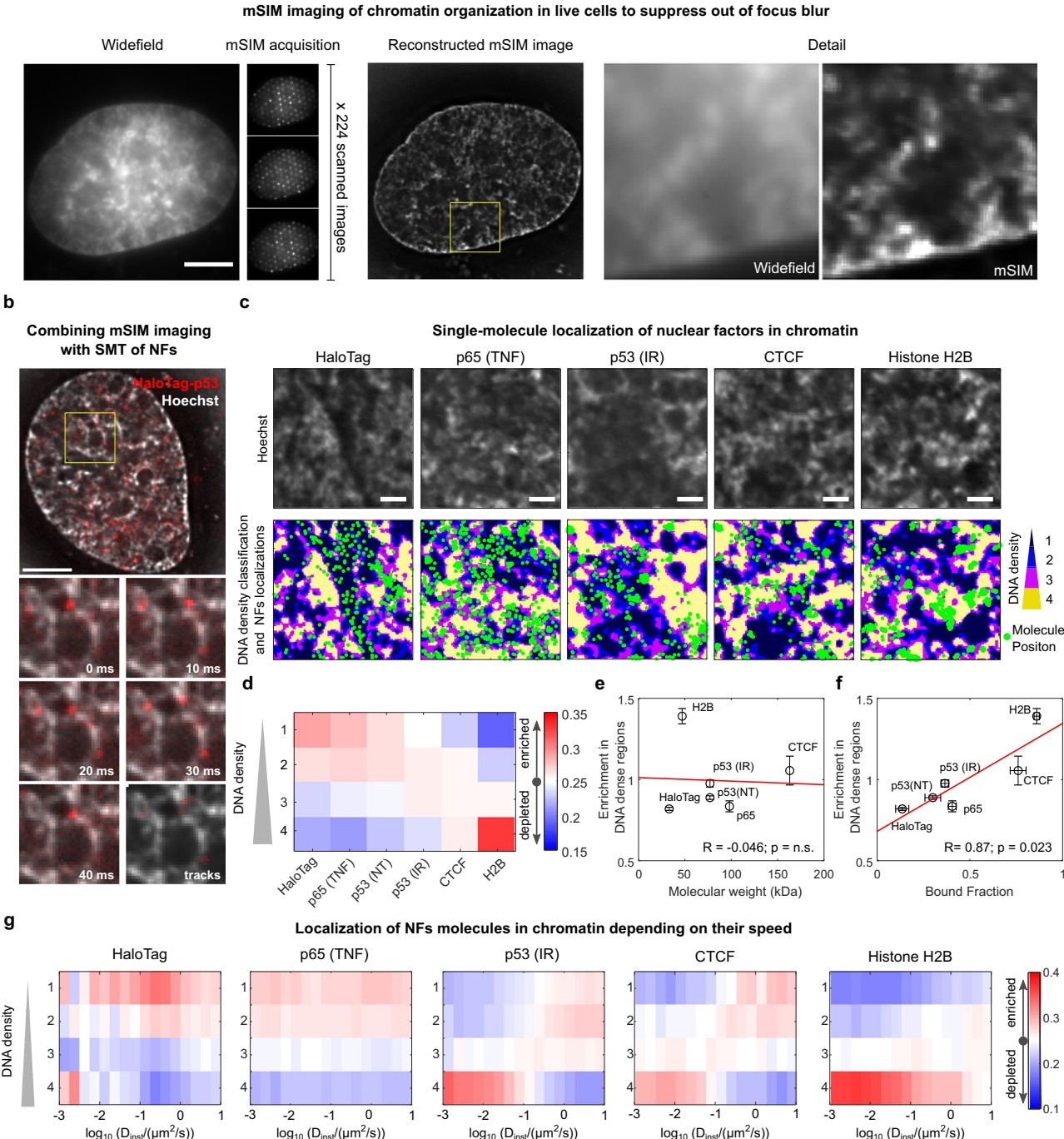

**Fig. 1 | SMT/mSIM microscopy probes factor-specific diffusion in chromatin.**
**a** mSIM imaging is achieved by scanning an array of diffraction-limited spots on the sample. Shown is an example of mSIM image of a live-cell nucleus labeled with Hoechst 33342, displaying increased optical sectioning compared with Widefield (scale bar: 5 μm). **b** Representative frames of a live-cell acquisition combining mSIM imaging of DNA density with SMT of HaloTag-p53 (scale bar: 5 μm).
**c** Representative localization of NF molecules on regions with different DNA density. p65 nuclear localization was analyzed upon stimulation by 10 ng/ml TNF. p53 nuclear localization was analyzed in untreated (NT) or upon 10 Gy of ionizing radiation (IR). The position of single NF molecules (green dots) is overlaid to the map of Hoechst intensity, classified in quartiles ($n_{replicates}$ = 2 biologically

independent experiments on at least 15 cells per nuclear factor per replicate; scale bar: 1 μm). **d** Different nuclear proteins are enriched in regions with different nuclear densities. Enrichment in DNA-dense regions does not significantly correlate with the NFs molecular weight (**e**), but it does correlate with their bound fraction, i.e., the fraction of immobile molecules (**f**) ($n_{replicates}$ = 2 biologically independent experiments, error bar: SD, statistical test: Pearson correlation). **g** Localization of NFs molecules in chromatin depending on their instantaneous diffusion coefficient ($n_{cells}$ = 31, 32, 29, 31, 32, from two biologically independent experiments for HaloTag, p65, p53, CTCF, and Histone H2B, respectively). Source data are provided as a Source Data file.

others preferentially occupy regions at defined DNA density, independently of their diffusion coefficient.

## Different NFs display distinct search strategies

Since our results indicated that different nuclear proteins display different mobility in chromatin, we next investigated how these proteins search for their binding sites and whether their search is influenced by chromatin organization. The search mechanism of NFs can be classified under two universal classes: non-compact exploration—where the factor has a high probability of leaving a certain volume before having explored it completely, e.g., in the case of Brownian diffusion—and compact exploration, where the protein exhaustively explores a given region before leaving it, as in the case of diffusion in fractal/crowded environments[8,9]. These different search strategies can be highlighted by analyzing the anisotropy of diffusion—i.e., the likelihood of unbound molecules to walk back on their steps—by calculating the distribution of angles between three consecutive localizations (Fig. 2). Brownian motion generates isotropic diffusion, while compact exploration would display backward anisotropy[8,9].

To compute diffusional anisotropy, we labeled the different nuclear proteins with a photoactivatable ligand (PA-JF$_{549}$)[32] and acquired photoactivated SMT (paSMT) movies[33] composed by 10,000 frames at 100 fps (Fig. 2a and Supplementary Movie 2) at a low molecule density (1–2 localized molecules per frame). Since the inclusion of bound molecules can bias the measurement of diffusional anisotropy, we filtered out those immobile/bound track segments using a Hidden Markov Model approach (variational Bayes single particle tracking, vbSPT)[34] and verified that the filtered tracks retained a minimal (<5%) bound fraction for all factors (Supplementary Fig. 2c).

We next measured diffusional anisotropy using the 'fold-anisotropy metric' $f_{180/0}$[19], describing how many times a molecule is more likely to show backward steps than forward steps (Fig. 2b). Notably, each NF displayed a different degree of diffusional anisotropy: factors showing poor localization in DNA-dense regions display low anisotropy (HaloTag and p65), compatible with non-compact exploration. Differently, factors enriched in DNA-dense regions such as CTCF and H2B showed higher diffusional anisotropy, a hallmark of more compact exploration (Fig. 2c).

Plotting the fold-anisotropy metric $f_{180/0}$ as function of the distance run by the molecules highlights that p53, CTCF, and H2B display high diffusional anisotropy at a spatial scale of 100–300 nm, and more isotropic diffusion when jumping longer distances (Fig. 2d). This behavior, recently named 'guided exploration'[19], occurs when a nuclear protein is trapped transiently in specific areas but is capable of diffusing freely between trapping zones, potentially accelerating its search for target sites. Since our data showed that diffusion of some NFs can be slowed down in DNA-dense nuclear areas, we reasoned that chromatin itself could be responsible for transiently trapping these factors. By combining the analysis of diffusional anisotropy with our mSIM imaging of DNA density, we observe that NFs display higher diffusional anisotropy when slowly diffusing in DNA-dense regions (Fig. 2e). Thus, factors like p53 and CTCF alternate between isotropic and fast diffusion in DNA-poor regions with slower and more anomalous diffusion in DNA-dense areas.

In summary, factors displaying different enrichment in DNA-dense regions display also different search strategies. Proteins with limited accessibility to dense DNA perform non-compact search in the nuclear environment, while the histone H2B—that preferentially occupies DNA-dense areas—shows the highest degree of compact exploration. CTCF and p53 perform guided exploration, potentially using DNA as a template: for these factors, slow-diffusing molecules are enriched in CD regions and appear trapped, while fast-diffusing molecules are enriched in regions at low DNA density and explore such areas in a non-compact manner.

## p53 searches its targets by a combination of fast and slow diffusion

We next focused on characterizing the search mechanism of the tumor-suppressor p53, an important TF controlling the cellular response to genotoxic stress. We genetically engineered the U2OS-derived DIvA cell line[35] to homozygously label endogenous p53 with HaloTag at its C-terminus, by means of CRISPR/Cas9[36]. These cells allow for enzymatic induction of double-strand breaks by inducing the translocation of the AsiSI restriction enzyme with 4-Hydroxytamoxifen (4-OHT). After verifying that p53-HaloTag is fully functional (Supplementary Fig. 3a–d), we applied paSMT to characterize p53 mobility before and after its activation by DNA damage, induced by ionizing radiation (IR) or AsiSI translocation. In agreement with previous reports[16,37], the distribution of frame-to-frame displacements for p53-HaloTag (Fig. 3a) is described by three apparent populations with diffusion coefficients given by $D_{bound} = 0.079 \pm 0.018 \, \mu m^2/s$, $D_{slow} = 1.06 \pm 0.24 \, \mu m^2/s$ and $D_{fast} = 5.1 \pm 0.7 \, \mu m^2/s$ in the untreated conditions and $D_{bound} = 0.068 \pm 0.017 \, \mu m^2/s$, $D_{slow} = 0.97 \pm 0.24 \, \mu m^2/s$ and $D_{fast} = 4.4 \pm 0.6 \, \mu m^2/s$, upon IR (Fig. 3b). As we previously demonstrated[16,37], the slowest of these three components represents DNA binding, since a p53 mutant with mutations at the seven residues responsible for specific DNA contacts (p53-mSB) displays a significant reduction in the fraction of molecules involved in that component (see below). Notably, four hours after DNA damage induction by 10 Gy of IR, the fraction of p53-HaloTag molecules bound to chromatin increases, indicating more efficient recruitment to binding sites (Fig. 3b), and confirming our previous results upon ectopic expression of p53-HaloTag in other cell lines[37]. The two diffusing components instead most likely represent 'effective' populations that group together multiple heterogeneous p53 behaviors. Nevertheless, model selection based on Bayesian Information Criteria, converges on this model (Supplementary Fig. 3e), and segmentation of diffusing tracks into 'slow' and 'fast' using vbSPT (Supplementary Fig. 3f) is useful to identify whether p53 molecules with different diffusion coefficients explore the nucleus and access chromatin differently.

After controlling that no misclassification of bound track segments in the diffusing ones was present (Supplementary Fig. 3f), we quantified diffusional anisotropy and found that the increased bound fraction of p53-HaloTag upon IR is associated with more pronounced compact exploration of its unbound population (Fig. 3c). Further, the transition probabilities derived from the vbSPT model highlighted that p53-HaloTag molecules reach the bound state ten times more frequently starting from the slow state than from the fast one (Fig. 3d). Accordingly, bound molecules more frequently co-cluster with slow-diffusing ones than with fast diffusing ones, as can be quantified by computing the pair cross-correlation between the localizations assigned to each of the states (Supplementary Fig. 3g). While the cross-correlation between bound and fast diffusing molecules closely resembles the one expected from randomly positioned molecules, slowly diffusing molecules spatially segregate at short distance from bound ones (Supplementary Fig. 3g). Together, these data indicate that p53 searches for its targets by alternating between faster and slower diffusion, with the slow-diffusing molecules performing more compact exploration and occupying more frequently nuclear regions that can be bound by the TF.

## Bound, slow, and fast p53 molecules are enriched at distinct DNA densities

Next, we combined p53 track classification with the maps of DNA density obtained by mSIM, to highlight how molecules with different mobility preferentially occupy regions at different DNA densities (Fig. 4a). By averaging the normalized distribution of Hoechst signal over thousands of classified p53 single molecules, we could visualize that fast diffusing molecules preferentially occupy DNA depleted regions, surrounded by regions at higher DNA density—such as the

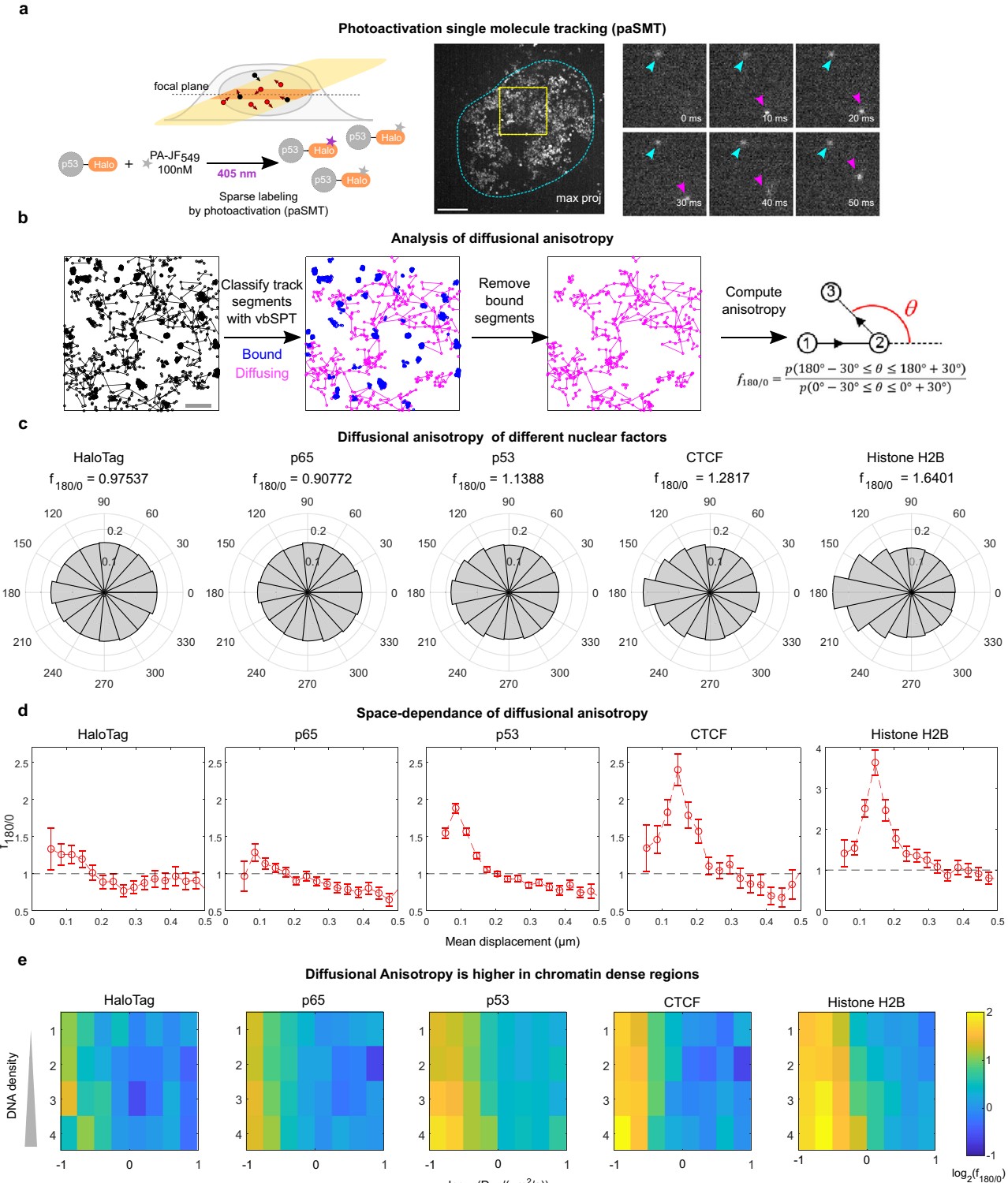

**Fig. 2 | Different NFs display different degrees of anisotropic diffusion. a** paSMT is carried out using highly inclined laminated optical sheet (HILO) microscopy (top left) by labelling the endogenous HaloTag-p53 with the photoactivatable dye PA-JF$_{549}$ (bottom left). Movies, acquired at a framerate of 100 fps highlight quasi-immobile chromatin-bound molecules (cyan arrowhead) and diffusing (purple arrowhead) ones (max proj = maximal projection over the entire movie; cyan dotted line indicates the cell nucleus, scale bar: 5 μm). **b** We use vbSPT to classify track segments into bound and diffusing components, and then focus on diffusing molecules, by computing diffusional anisotropy, by calculating the distributions of angles θ between consecutive jumps, and the fold-anisotropy metric, $f_{180/0}$, calculated as the probability of observing a backward displacement $p(150° ≤ θ ≤ 210°)$ over the probability of observing a forward displacement $p(−30° ≤ θ ≤ 30°)$. **c** Different NFs display different

diffusional anisotropy, with factors poorly localized in DNA-dense regions displaying lower anisotropy than factors enriched in DNA-dense regions. **d** Fold-anisotropy metric $f_{180/0}$ as function of the distance run by the molecules. p53, CTCF, and H2B display high diffusional anisotropy at a spatial scale of ~100–150 nm, a signature of transient trapping of these molecules in traps of similar size ($n_{cells}$ = 30, 30, 29, 14, 31, $n_{angles}$ = 59470, 62813, 180414, 26052, 26566 for HaloTag, p565, p53, CTCF, and Histone H2B respectively, error bars: s.e.m. estimated through boot-strapping). **e** Analysis of diffusional anisotropy in our SMT/mSIM data allows us to identify that the highest diffusional anisotropy occurs for molecules with slow instantaneous diffusion coefficients in regions at high chromatin density (same data as in Fig. 1c–g). Source data are provided as a Source Data file.

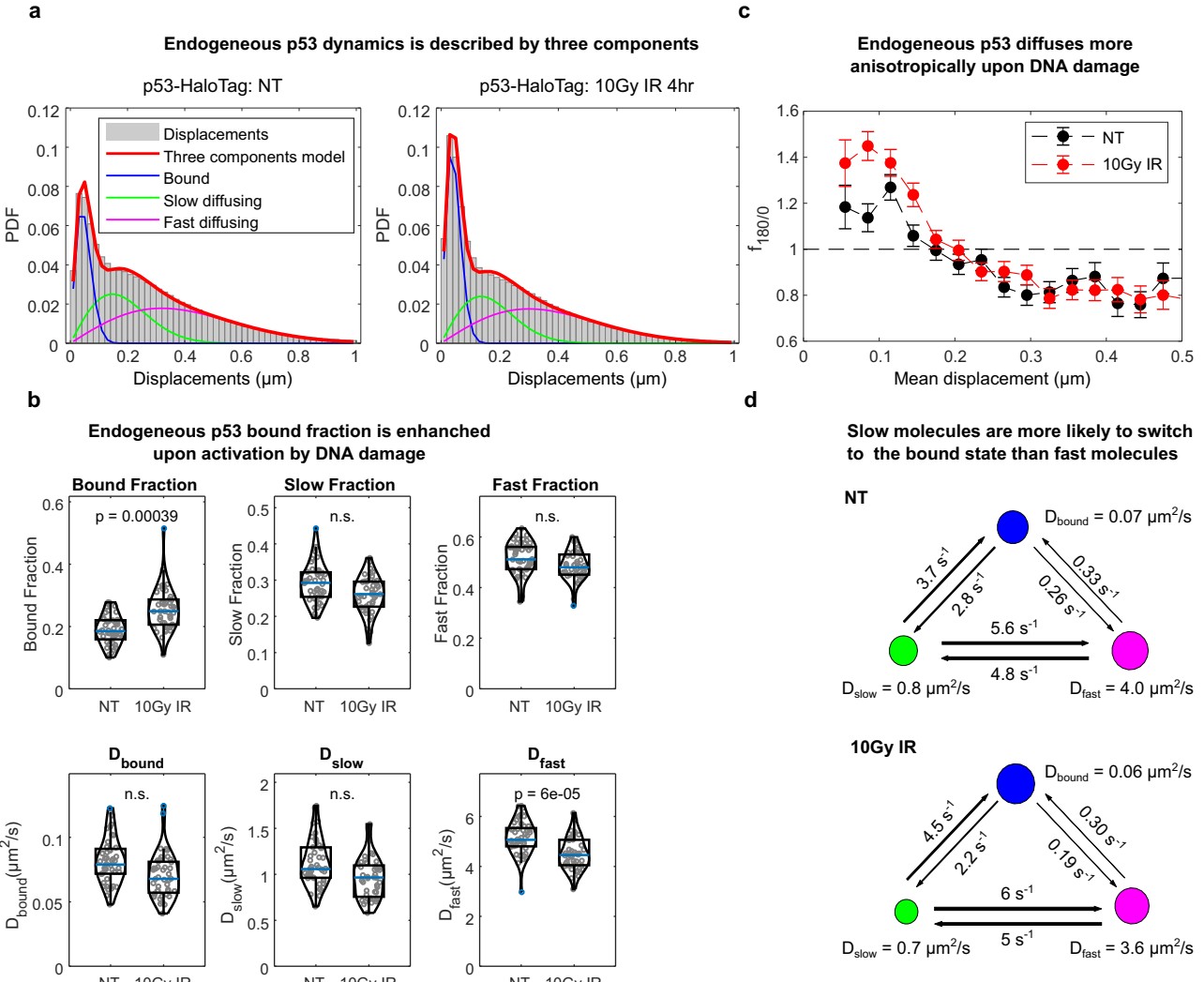

**Fig. 3 | Single-molecule diffusion of endogenously tagged p53 in DIvA cells.**
**a** The distribution of single-molecule displacements before and after the induction of DNA damage via IR shows that p53 is described by a model accounting for three populations (red line, Eq. 2 in the Methods section), a bound one (blue line) and two diffusing ones (green and magenta lines). **b** The parameters extracted by fitting the distribution of displacements show that upon activation of p53 the p53 bound fraction increases, while p53 diffusion is slowed down (the blue line represents the median, box edges represent upper and lower quartiles and whiskers extend between Q1–1.5 IQR and Q3 + 1.5 IQR, where IQR is the interquartile range,

$n_{cells}$ = 45, 44 for untreated and 10 Gy IR respectively, statistical test: two-sided Kolmogorov–Smirnov, with Bonferroni correction for multiple testing). **c** Diffusional anisotropy profile of endogenous p53-HaloTag is compatible with guided exploration $n_{cells}$ = 45, 44, $n_{angles}$ 1269065, 151712, for NT and IR respectively −error bars: s.e.m. estimated through boot-strapping). **d** The classification of track segments using vbSPT allow to compute the transition probability between the bound state and the two diffusing states. Both before and after activation by IR, p53 molecules are 10× more likely to perform the slow-to-bound transition than the fast-to-bound one. Source data are provided as a Source Data file.

channels composing the IC—while bound molecules are enriched in denser chromatin domains (CDs), surrounded by DNA poor regions (Fig. 4b)[5]. Slow-diffusing molecules localize at regions of intermediate DNA density (Fig. 4b) that might include regions at the IC/CD boundary previously termed the 'perichromatin' compartment[5]. Activation of p53 by 10-Gy IR further promotes the enrichment of p53 bound molecules in DNA-dense compartments (Fig. 4b). Notably, activation of p53 by AsiSI induction of DNA damage recapitulates the previous observations: in these settings p53-HaloTag displays an increased bound fraction (Supplementary Fig. 4a), a diffusion anisotropy profile compatible with guided exploration (Supplementary Fig. 4b) and positioning of bound p53 molecules at regions of higher DNA density than diffusing ones (Supplementary Fig. 4c, d).

Since p53-bound molecules preferentially reside in DNA-dense domains, we asked whether p53 target genes occupy regions of similar density: we performed a DNA FISH experiment on the *CDKN1A* gene locus, measuring its position relative to chromatin (Fig. 4c).

Interestingly, the *CDKN1A* locus is preferentially positioned at dense DNA sites, surrounded by regions at lower compaction, in agreement with our hypothesis (Fig. 4d).

Thus, p53 scans the nuclear environment by diffusing fast and isotropically in the IC and slowing down, performing more compact exploration of regions at higher DNA compaction, where the *CDKN1A* locus is preferentially positioned.

### The p53 search depends on its DNA binding domain and IDRs
To dissect the mechanism controlling the DNA density-guided search process of p53, we analyzed the diffusional behavior of HaloTag-p53 mutants, ectopically expressed in a p53 KO DIvA cell line, generated by CRISPR-Cas9 mediated gene editing. Previous reports indicate that the IDRs of NFs might be responsible for their target search[19,38,39]. Analysis of the p53 disorder reveals that both the N-terminus and the C-terminus of p53 WT are disordered (Fig. 5a). We generated HaloTag-p53 mutants lacking either the N-terminus (p53ΔN, lacking both its

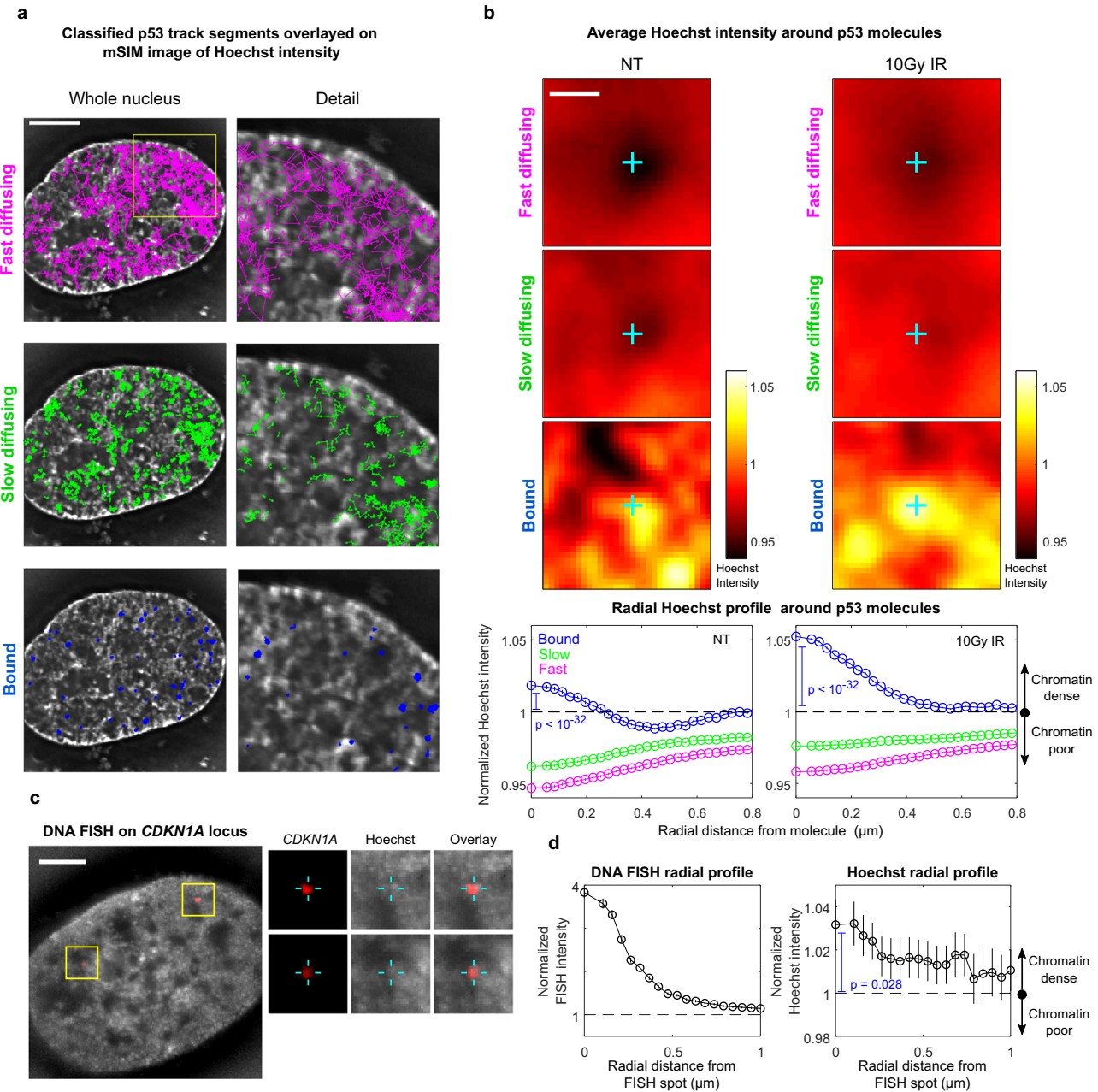

**Fig. 4 | SMT/mSIM experiments highlight that p53 molecules with different diffusivity preferentially occupy regions with different DNA density.**
**a** Classified p53 tracks displayed over the mSIM image of DNA density ($n_{replicates}$ = 2 biologically independent experiments on at least 15 cells per condition per replicate; scale bar: 5 μm). **b** Average Hoechst intensity (top, the cyan cross represents the position of the p53 molecule) and average radial Hoechst profile (bottom) around bound, slow-diffusing, and fast-diffusing p53 molecules. The images and the radial profiles are normalized to the average nuclear Hoechst intensity: values higher than 1 represent regions denser than average, and values lower than one represent chromatin-poor regions. Bound molecules are enriched at regions at DNA density higher than average surrounded by chromatin regions at lower density, while free molecules are enriched in chromatin-poor regions surrounded by denser ones. Slow-diffusing molecules displaying an intermediate profile. DNA damage further increases enrichment of bound molecules at DNA-dense regions (error bars: s.e.m. $n_{cells}$ = 29, 29 for untreated and 10 Gy IR, respectively,

$n_{molecules}$ = (90137,99583,70877),(174163,153493,90949) for untreated (bound, slow, fast) and 10 Gy IR (bound, slow, fast) respectively scale bar: 0.5 μm, statistical test: two-sided, one-sample $t$ test to probe that the first point of the profile is significantly higher than 1, i.e., bound molecules sit at higher chromatin density than average). **c** mSIM acquisition of the chromatin surrounding the *CDKN1A* locus (red: *CDKN1A* locus, grey: Hoechst 33342, scale bar: 5 μm. $n_{replicates}$ = 2 biologically independent experiments on at least 20 cells per replicate. Inset shows the localization of *CDKN1A* spots). **d** We averaged together 2 μm ROIs centered around 53 DNA FISH spots and computed the average radial profile of DNA density. On average, the *CDKN1A* locus appears to be positioned in regions with higher chromatin density than its surroundings (error bars: s.e.m, statistical test: two-sided one-sample $t$ test to probe that the first point of the profile is significantly higher than 1, i.e., the *CDKN1A* locus sits at higher chromatin density than average). Source data are provided as a Source Data file.

transcriptional activation domain, TAD and the proline-rich domain, PRD) or the C-terminus (p53ΔC, lacking the last 30aa of the protein). Notably, p53ΔC has been shown to be incapable of non-specific sliding in vitro[29]. On the other hand, our initial analysis of different NFs

displayed an interesting correlation between the ability of factors to be enriched in DNA-dense regions and their bound fraction, which might be a proxy for their affinity to chromatin. We, therefore, included in our analysis the p53-mSB mutant that is incapable of specific binding

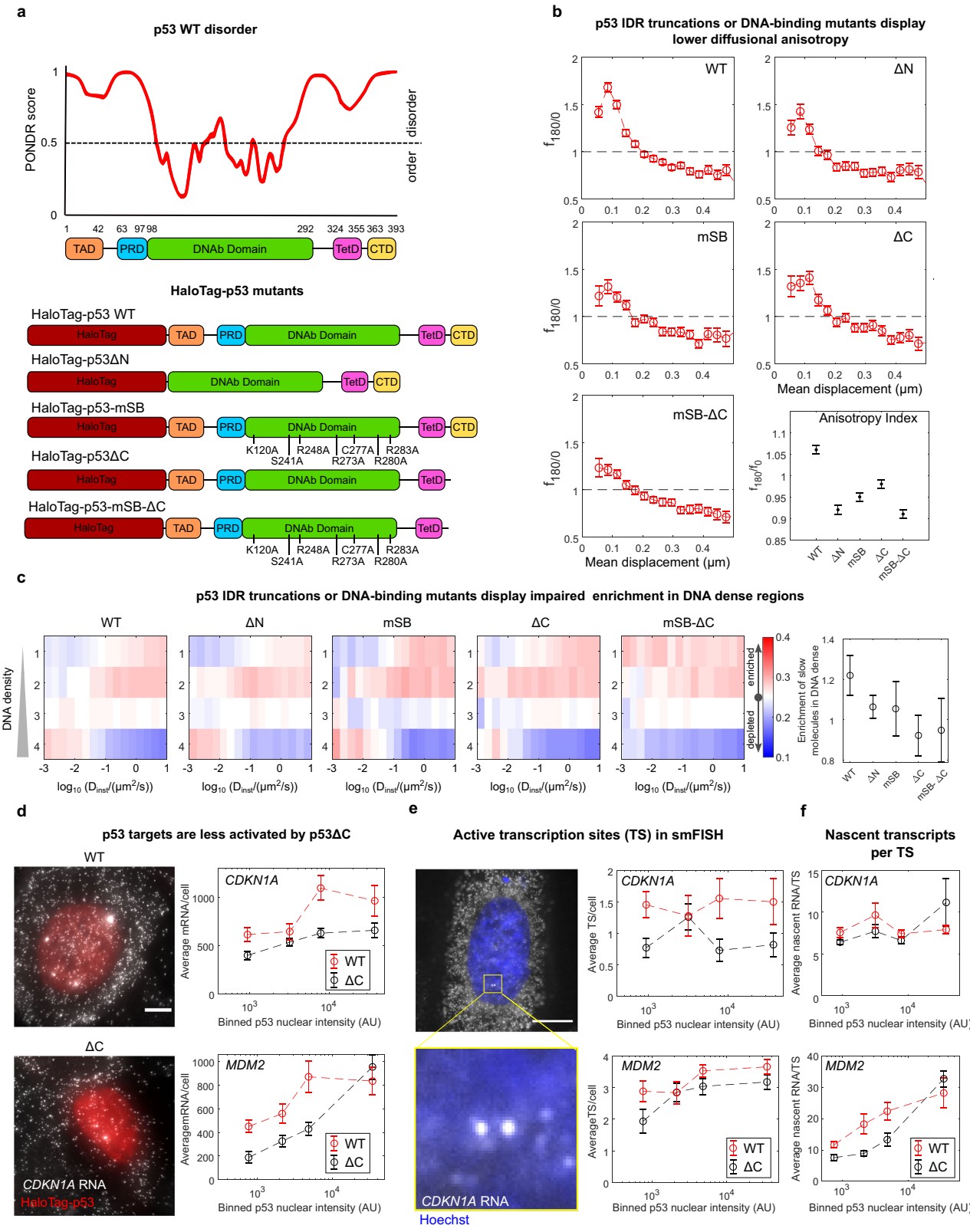

on DNA, given that all the seven amino acids responsible for specific contacts with DNA are mutated to alanine[40]. Finally, a mutant lacking both specific binding and its C-terminus region was included (p53-mSB-ΔC). We collected paSMT movies on these mutants, and analyzed their single-molecule dynamics (Supplementary Fig. 5a). Mutants involving the DNA binding domain and/or the C-terminal region displayed a significant reduction in the TF bound fraction, associated with

an increase in the fraction of diffusing molecules (Supplementary Fig. 5b). Further, all tested mutants displayed a reduced backward diffusional anisotropy compared to p53 WT, indicating that both the IDRs and the DNA binding domain of p53 might participate in guiding the TF to its binding sites (Fig. 5b). Next, we resorted to SMT/mSIM to evaluate the enrichment of these mutants at DNA-dense regions of the nucleus. While p53 WT displays an enrichment of bound/slow-

**Fig. 5 | The p53 search mechanism is governed by interactions mediated by its IDRs and DNAb domains. a** p53 disorder evaluated by PONDR (Top) and list of probed p53 mutants. **b** All mutants display reduced diffusional anisotropy compared to p53 WT ($n_{cells}$ = 45,30,28,24,31, $n_{angles}$ = 262907, 132280,124489, 123006, 153235 for WT, ΔN, mSB, ΔC, mSB-ΔC, respectively, error bars: s.e.m. estimated through bootstrapping). **c** Localization of mutant-p53 molecules by SMT/mSIM, shows that the p53 mutants lacking the C-terminal IDR display impaired recruitment to chromatin-dense regions ($n_{cells}$ = 42,34,20,22,9 from two biologically independent experiments for WT, ΔN, mSB, ΔC, mSB-ΔC, respectively, scatter plot: mean±SD). **d** p53 target gene expression as function of HaloTag-p53 levels analyzed

by smFISH. Shown is maximal projection of 3D stack (left, scale bar: 5 μm) and Average mRNA counts for two p53 targets, in cells expressing either HaloTag-p53 WT or HaloTag-p53ΔC, as a function of HaloTag-p53 levels (right, $n_{cells}$ = 74,104,50,88, for WT-*CDKN1A*, ΔC-*CDKN1A*, WT-*MDM2*, ΔC-*MDM2* respectively, error bars: s.e.m.). smFISH allows to estimate the number of active transcription sites (TS) per nucleus (**e**) and the number of nascent transcripts per TS (**f**) (scale bar: 5 μm, $n_{cells}$ = 74,104,50,88, for WT-*CDKN1A*, ΔC-*CDKN1A*, WT-*MDM2*, ΔC-*MDM2* respectively, error bars: s.e.m.). Source data are provided as a Source Data file.

diffusing molecules in regions at high-DNA density, this enrichment is progressively lost, in particular when analyzing the localization of mutants lacking the C-terminal domain of the protein (Fig. 5c). Thus, in agreement with our previous results on different NFs, those p53 mutants displaying lower bound fraction (such as IDR lacking mutants) are also less capable of being targeted to regions at higher DNA density.

If, indeed, deletion of p53 IDRs perturbs the targeting of the TF to its binding sites, one would expect these mutants to display a lower capability to induce transcription. To test this possibility, we compared the expression of two p53 targets, *CDKN1A* and *MDM2*, in cells expressing p53 WT, or p53ΔC (the only mutant displaying transcriptional activity in luciferase assays[41]) over a p53-null background. To account for the cell-to-cell heterogeneity of ectopically expressed p53, we used single-molecule fluorescence in situ hybridization (smFISH), to quantify the number of expressed mRNAs at the single cell level and to relate it with p53 abundance (Fig. 5d). Notably, at similar protein expression levels, the p53ΔC mutant displayed lower mRNA counts than p53 WT (Fig. 5d). Such difference across p53 mutants was also reflected at the nascent transcription level, by quantifying the number of transcription sites (TS) per cell, that appear as 1 to 4 bright foci in the nucleus (Fig. 5e). Finally, we quantified the number of nascent transcripts per TS (Fig. 5f). Higher number of nascent RNA per TS can be a consequence of both higher transcriptional burst amplitude (when transcription initiation is infrequent) and of increased frequency of transcriptional initiation (when transcription initiation is more frequent than the rate at which mRNA is released from the TS[42]). The *CDKN1A* gene displays a constant number of nascent RNAs per TS with increasing levels of p53 WT or p53ΔC. Thus, both factors are capable of activating transcription to similar extents once they reach the *CDKN1A* locus, but p53 WT appears to reach it more frequently than p53ΔC. For *MDM2* instead higher p53 concentration stimulates a higher number of *MDM2* nascent RNAs produced per transcription site, potentially by increasing the frequency at which *MDM2* transcription is initiated within a burst. Yet, we note that at any given concentration, p53ΔC generates less *MDM2* nascent RNAs than p53 WT. These results are therefore consistent with the hypothesis that the p53ΔC might be less efficiently targeted to its genes, reducing their expression both at the mature and nascent RNA levels.

**An IDR fused to p53 can modulate and divert the p53 search**
Since deletion of the p53 IDRs resulted in a reduction of the protein compact exploration signature, we aimed to investigate the effect of adding an exogenous IDR to p53 on the TF search mechanism.

To this purpose, we fused the IDR derived from FUS protein, to the full p53 coding sequence (Fig. 6a). The HaloTag-FUS-p53 construct displayed nuclear localization and, differently from p53 WT, the capability of forming condensates in living cells at expression levels higher than a certain nuclear level (located around the 60-th percentile of our transfected cell population, Fig. 6b, c). Analysis by Fluorescence Recovery After Photobleaching (FRAP) revealed that these condensates display 'solid' properties, with a high immobile fraction (Supplementary Fig. 6). Notably, even outside of these condensates (where a large proportion of HaloTag-p53 still resides), FUS-p53

diffuses slower than p53 WT (Supplementary Fig. 6). Accordingly, analysis of mobility at the single-molecule level by paSMT, highlights that FUS-p53 molecules diffuse with lower diffusion coefficients than p53 WT and a larger fraction of its molecules are classified as bound (Fig. 6d). Diffusing FUS-p53 also displayed a stronger signature of guided exploration with a more prominent anisotropy peak at spatial scales below 150 nm (Fig. 6e). Thus, FUS-p53 samples the nuclear environment in a more compact manner, further highlighting that TFs IDRs play a major role in determining the search mechanism employed by the TF. We next employed the SMT/mSIM combined approach to monitor how this more compact exploration affects the capability of targeting FUS-p53 to DNA-dense domains. To our surprise, however, the analysis revealed that—on average—FUS-p53 is not enriched at DNA-dense regions, independently of its diffusion coefficient (Fig. 6f).

We thus applied smFISH to probe the effect of the exogenous IDR on the transcription of *CDKN1A* and *MDM2*, at the single-cell level. p53 WT and FUS-p53 resulted in remarkable differences in the transcriptional output as a function of the TF expression levels. Transcripts of *CDKN1A* and *MDM2* were found to increase with p53 WT levels, as expected for a transcriptional activator (Fig. 6g). The amount of synthesized mRNA in response to FUS-p53 displayed instead a more peculiar profile: at expression levels lower than a nuclear level threshold, which is found between the 50% and the 67% percentile of our cell population, FUS-p53 resulted in a higher expression of *CDKN1A* and *MDM2* mRNA (up to 3–4-fold) than p53 WT (Fig. 6g). This more rapid increase was also observed for the number of active TS observed per cell, suggesting that in this regime of low FUS-p53 expression levels, the targeting of the activator to its regulated genes is enhanced. At FUS-p53 expression levels higher than this threshold, we instead observed a decrease in mRNA counts and active TS per cell (Fig. 6g). This observation was reproduced in another cellular model (p53 KO MCF7 cells[37], Supplementary Fig. 7a), while no modulation in expression depending on FUS-p53 levels was observed for the p53-unrelated house-keeping gene *GAPDH* (Supplementary Fig. 7b). Of note, the FUS-p53 levels at which transcription starts to be inhibited roughly matches with the one at which visible FUS-p53 condensates appear (Fig. 6c). This data suggest that IDR-mediated interactions need to be finely regulated in order to promote the targeting of the TF on its binding sites, as an excess of IDR-mediated interactions could divert the search mechanism and inhibit transcriptional activation. To test this hypothesis, we therefore re-analyzed our SMT/SIM acquisition on FUS-p53 by splitting the dataset in two subgroups: cells expressing FUS-p53 at low levels, with no visible condensation and cells expressing FUS-p53 at high levels and displaying two or more condensates (Fig. 6h). Such analysis highlighted that bound/slow-diffusing FUS-p53 can be targeted to regions at higher DNA density when expressed at low levels, while high expression of FUS-p53 derails this targeting, resulting in the localization of both diffusing and bound p53 molecules in chromatin-poor regions.

In summary, our data demonstrate that DNA density modulates the target search of p53, through interactions with the protein DNA binding domain and its IDRs, and that these interactions need to be finely regulated since both IDR deletions or potentiation can have

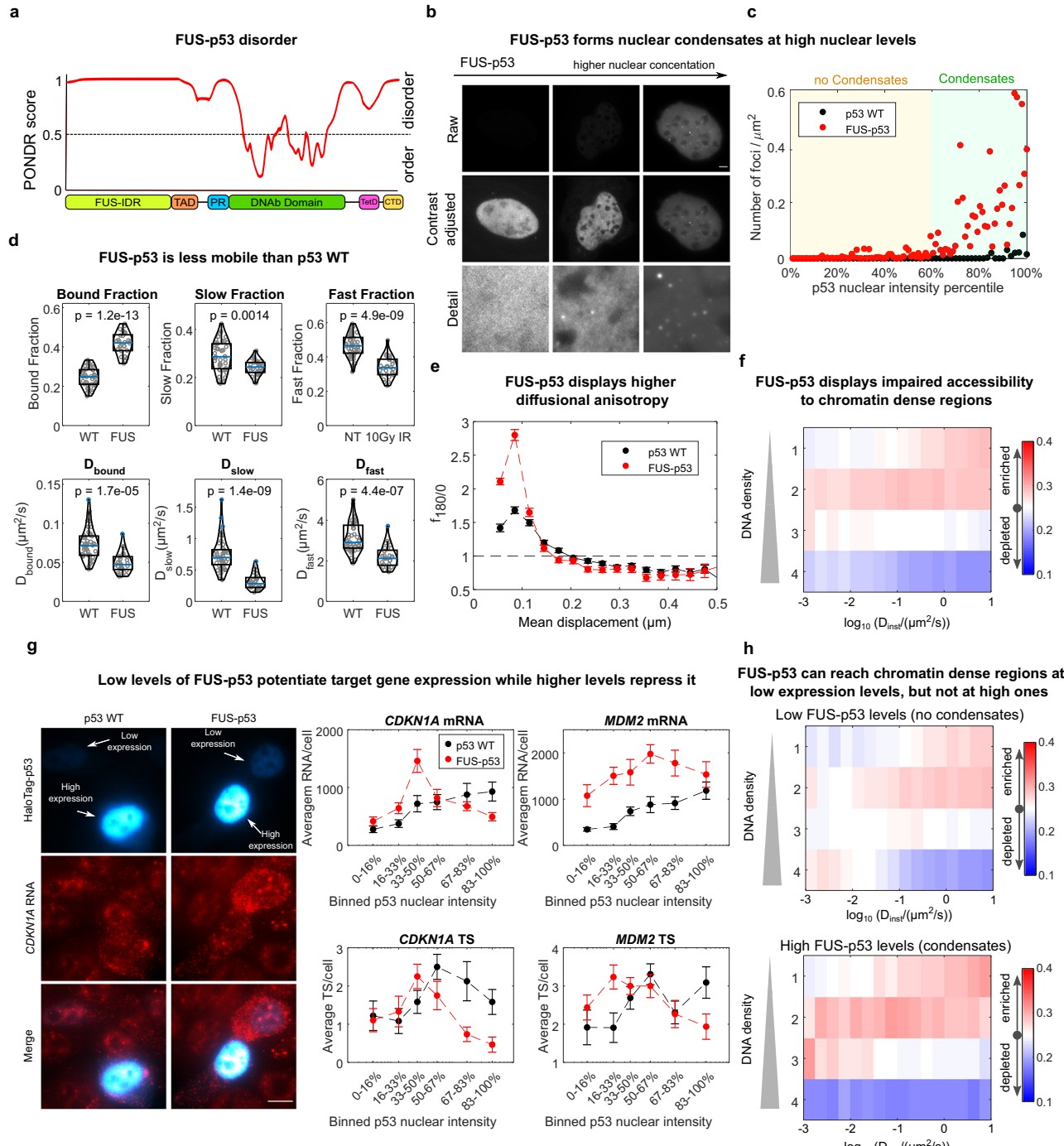

**Fig. 6 | Fusing an exogenous IDR to p53 renders its diffusion more compact, but interferes with its targeting to DNA-dense regions. a** PONDR analysis of the FUS-p53 construct. **b** FUS-p53 forms intranuclear condensates above a certain nuclear level that can be estimated **c** approximately at the 60th percentile of our transfected cell population ($n_{cells}$ = 60, 96 for p53 WT and FUS-p53, experiment repeated on two independent replicates; scale bar: 5 μm). **d** paSMT shows that FUS-p53 displays a higher fraction of molecules in bound state and slower diffusion coefficients (the blue line represents the median, box edges represent upper and lower quartiles and whiskers extend between Q1–1.5 IQR and Q3 + 1.5 IQR, where IQR is the interquartile range, ($n_{cells}$ = 45, 29 for p53 WT and FUS-p53 respectively, statistical test two-sided Kolmogorov-Smirnov, with Bonferroni correction for multiple testing). **e** FUS-p53 also displays a more prominent diffusional anisotropy peak ($n_{cells}$ = 45, 29 for p53 WT and FUS-p53 respectively, error bars: s.e.m. estimated by

bootstrapping). **f** SMT/mSIM reveals that FUS-p53 accessibility to DNA-dense regions is impaired. **g** Simultaneous imaging of HaloTag-p53 nuclear levels and mRNA expression by smFISH (False colors are used in the HaloTag-p53 channels to visualize both low expressing and high expressing cells, scale bar: 5 μm) highlights that p53 target genes are activated in a biphasic manner by FUS-p53. At low nuclear levels, FUS-p53 activates target genes more efficiently than p53 WT, while at higher expression levels FUS-p53 is more repressive ($n_{cells}$ = 65, 102, 77, 93 for WT-*CDKN1A*, FUS-*CDKN1A*, WT-*MDM2*, FUS-*MDM2* samples respectively, error bars: s.e.m.). **h** Stratifying FUS-p53 mSIM/SMT data in cells without and with visible clusters highlights that FUS-p53 binds regions at higher chromatin density when expressed at low levels, with no visible clusters. ($n_{cells}$ = 22,13 for "no-condensate" and "with condensates" datasets respectively). Source data are provided as a Source Data file.

detrimental effects on p53 targeting and on transcription of its target genes.

## Discussion

How TFs scan the genome to identify the subset of their responsive elements to bind and regulate, and what the role of chromatin organization in directing this search in eukaryotes are fundamental unanswered questions. To address these questions, we have developed a microscope that allows overlaying single-molecule tracks of TFs on maps of chromatin organization obtained by structured illumination. We used the microscope to characterize how DNA density modulates the search process of different NFs, with particular attention to the search mechanism of the tumor-suppressor p53.

Our data show that individual NFs display distinct accessibility to DNA-dense regions. Some factors are excluded from CD regions and display isotropic diffusion (the inert tracer HaloTag) or low anisotropy (p65). Others, such as p53 and CTCF, display a specific enrichment of slow, anisotropically diffusing molecules in DNA-dense nuclear regions, suggesting that these factors might be trapped in these DNA-rich areas, to search for their target sites.

Part of this anisotropic diffusion might be due to molecules stably bound to mobile DNA rather than the NFs search. However, genomic loci typically display diffusion coefficients much lower[43] than $0.1\,\mu m^2/s$, while our anisotropy analysis is performed on the diffusing population of the factors, with diffusion coefficients of the order of 0.5 to $5\,\mu m^2/s$, and it thus predominantly reports on search properties rather than on chromatin mobility.

Our characterization of p53 diffusion at endogenous levels, revealed that the search mechanism of this TF is compatible with a guided exploration mechanism. Specifically, p53 search can be best described by two effective populations, slow and fast. These 'slow-diffusing' and 'fast-diffusing' molecules likely represent heterogeneous populations, possibly characterized by a spectrum of diffusion coefficients[16]. Yet, such classification allows us to relate more clearly p53 mobility, p53 search strategy, and its accessibility to DNA-dense regions. Indeed, p53 alternates between fast, non-compact exploration and slower and more compact diffusion where molecules sample ~100 nm sized domains exhaustively by frequently walking back on their steps, as evidenced by p53 anisotropic diffusion at short spatial scales. These slow and compact diffusing molecules scan and interrogate the genome to identify their target sites, since they frequently co-localize with bound molecules. Such guided exploration appears to be modulated upon p53 activation by DNA damage and can speed up the target search, as a CTD deletion mutant with reduced anisotropic diffusion displayed reduced efficiency in the activation of target genes.

Guided exploration is emerging as a common mechanism to accelerate the search process of nuclear proteins: for example, the chromatin factors CTCF[19] and the Polycomb subunit CBX2[20] display similar compact diffusion profiles to the ones measured here for p53. However, while CTCF forms nuclear clusters[44] and CBX2 forms nuclear condensates that act as trapping zones for the guided exploration mechanism, p53 WT does not appear to be involved in the formation of visible clusters.

Our combined SMT/SIM approach points instead to chromatin itself as the template for p53 compact exploration. p53 fast diffusing molecules are preferentially localized in DNA-poor nuclear regions (i.e., the IC) while slow-scanning molecules appear to diffuse on regions at higher DNA density—potentially representing the surface of CD regions—and bound molecules often localize inside or at the edge of these CD domains. This result might reflect the behavior of p53 as a pioneer factor that binds responsive elements in inaccessible CD regions[45,46], with recent reports highlighting that p53 has pioneer activity only at nucleosome-dense regions flanked by nucleosome-poor ones[47,48].

The mutant analysis further suggested that p53-guided exploration is controlled by both its DNA-binding domain and IDRs, as a mutant with altered residues for DNA contacts and those lacking IDRs show impaired exploration and mis-localization relative to chromatin. In vitro studies have proposed that both the DNA-binding domain and C-terminal IDR might contribute to p53 search by scanning DNA regions[29,49]. Our findings now provide evidence of their role in p53 search in living cells and extend this observation to the p53 N-terminal IDR. This expands the current understanding that IDRs can influence the mobility of TFs and, in turn, their targeting process. In agreement with our observations on p53, SMT has been recently used to show that the Hypoxia-Inducible Factor (HIF) 1-α and HIF 2-α display interchangeable nuclear mobility when swapping their IDRs[50]. Importantly, mutants of the yeast TFs Msn2 and Yap lacking the DNA binding domain, can still locate most of their target genes through their IDRs[38].

In recent years, TF IDRs have drawn substantial attention also related to the capability of disordered proteins to form condensates that might enhance transcription by increasing the local concentration of the TF[23]. Recent data highlight that IDRs might be important for the transactivation potential of TFs independently on their capability to form condensates[26] and overexpression of an isolated IDR that can interact with specific TFs can divert the TFs search[25]. Our results extend these observations. Fusing p53 to an exogenous IDR (derived from FUS) increases compact exploration and the transcription of target genes when expressed at low levels. However, higher expression of FUS-p53 leads to the formation of solid FUS-p53 condensates and results in the derailment of the search process. In these conditions, FUS-p53 bound molecules do not localize any longer in DNA-dense regions and we observe downregulation of target gene transcription.

These results point to a model in which IDR-mediated interactions with the eukaryotic nucleus need to be finely balanced to warrant efficient targeting. According to this, p53 mutants deficient in IDR-mediated interactions explore the nuclear environment in a less compact manner and are less likely to find their binding sites, but adding exogenous IDR results in an excessively compact search that sequesters the TF away from its responsive elements. A similar trade-off has been described for facilitated diffusion, the most studied accelerated search mechanism in prokaryotes[12-14]. Here, optimal search is achieved when the factor spends 50% of the search time by sliding on DNA, as shorter or longer sliding times result in inefficient targeting[51].

Such model raises some novel and important questions that we hope to be capable of addressing in the near future like identifying the mechanism that controls p53-guided exploration at DNA-dense region: an interesting candidate that might mediate p53 compact exploration at CD edges is nascent RNA (as IDR/RNA interactions seem to control the guided exploration of other NFs[19], and the CD edges have been shown to be enriched in nascent RNA[7]) or protein/protein interactions (as for example active histone marks, general TFs, and pre-initiation complex members are found to be enriched at the boundary of CDs[7]). Another intriguing possibility is that chromatin dynamics itself could affect the search and the distribution of NFs in chromatin. Live-cell imaging studies have revealed spontaneous nucleosome fluctuation in mammalian nuclei. Such local movements could facilitate chromatin accessibility for various NFs, including proteins involved in DNA repair, as well as TFs searching for their specific targets[52,53]. On the other hand, one could speculate that these chromatin movements might also facilitate the transfer of target sequences to the surface of CDs, determining whether the target will be found and transcribed and contributing to the target selectivity of the TF. Further improvement of our SMT/mSIM approach to allow for the simultaneous imaging of NFs single-molecule mobility and fast chromatin movement at high resolution could help clarify the impact of chromatin dynamics on the NFs search mechanism.

Another important question is whether the search mechanism described here for p53 also applies to other TFs. On one side, the CD/IC interface appears as a peculiar compartment, enriched with nascent RNAs and permissive histone marks[7], and it appears therefore possible that many other factors would be targeted there with high efficiency. To a limited degree, however, data on other NFs seem to point against a common shared search mechanism. For example, the nuclear kinase P-TEFb shows compact exploration at all spatial scales[17], whereas factors such as the oncogene cMyc[17] and the NF-kB subunit p65 (this work) display a non-compact search. Interestingly, the low anisotropy index shown here by p65 (~0.9) might indicate directed motion. To our knowledge, no NF has been reported to undergo directed motion, except for the molecular motor Myosin VI[54], suggested to regulate the spatial organization of transcription[55]. Finally, other NFs display guided exploration that depends on the capability of the factor to form clusters[19,20]. Different TFs are likely to explore the nuclear environment differently, by transiently interacting with different nuclear structures, organelles/compartments, and molecular players. We envision that by increasing the throughput of the SMT/mSIM approach described here for probing multiple nuclear organelles, it will be possible to map the specific nuclear landscape sensed by any individual TF.

## Methods

### Cell lines

DIvA cells (originated from the human osteosarcoma U2OS) were a kind gift from the Gaëlle Legube lab[35]. The U2OS-derived line with endogenous expression of CTCF-HaloTag (used by us for control experiments), was a kind gift from the Tjian and Darzacq group[44]. MCF7 p53 KO cells were previously generated in our lab[37].

We used CRISPR/Cas9 gene editing to create p53-HaloTag knock-in (KI) and p53 knock-out (KO) cell lines. p53 KI DIvA cells were generated using four plasmids encoding the Cas9-D10A nickase[56], two guide RNAs (sgRNA1 and sgRNA2), and a repair vector carrying both the HaloTag gene and a neomycin resistance gene, flanked by two p53 homology arms (~800 bp). The sequences targeted by sgRNA1 and sgRNA2 on the p53 genomic locus were 5′-GATGACATCACATGAGTGAG-3′ and 5′-CAGCCACCTGAAGTCCAAAA-3′, respectively. DIvA cells seeded on a 6-well plate were co-transfected with the four plasmids (625 ng of each vector per well) using Lipofectamine 3000 (Thermo-Fisher, cat. L3000008) and following the manufacturer's protocol. After transfection, cells were left to recover a few days before two rounds of antibiotic selection (mild, 4 days and long, ~15 days) with geneticin (G418 Sulfate, Thermo-Fisher, cat. 10131027, used at 800 µg/mL). Cells were next seeded at single cell per well (96 well plate) to isolate single clones. The KI clone used in this study was functionally tested for nuclear localization and abundance of p53-HaloTag, for the absence of untagged p53, and for the capability to induce canonical p53 target genes (both at the RNA and protein level) (Supplementary Fig. 2).

To generate p53 KO DIvA cells, a commercial CRISPR/Cas9-based gene editing kit was employed (Santa Cruz Biotechnology, cat. sc-416469-NIC) including constructs expressing Cas9-D10A nickase, two sgRNAs and a GFP marker transiently expressed for selection. DIvA cells were transfected with UltraCruz Transfection Reagent (sc-395739) following the manufacturer's instructions and sorted for GFP the next day. Single clones were isolated and expanded from the sorted population. Successful knock-out of the p53-null clone employed in this work was confirmed by the absence of p53 and inactivation of p53 target genes (both at the RNA and protein level) (Supplementary Fig. 2).

### Plasmids and transient transfections

Plasmids expressing HaloTag-p53 mutants (deletions of the p53 TAD, CTD, or both; insertion of the FUS N-terminus domain) were synthesized and sequenced by the Genewiz Company from a pFN22A vector

(Promega) containing the insert HaloTag-p53 WT. The vectors expressing other HaloTag-conjugated proteins were: H2B (pFC15A backbone, Promega)[16], the NF-κB subunit p65 (pci-Neo-p65 HaloTag)[57], and HaloTag itself (pHTN, Promega).

P53-null cells were transiently transfected with plasmids. Briefly, DIvA p53 KO cells were seeded at ~30% of confluence, and 24 h later (~60% confluence) were transfected with Lipofectamine 3000 (Thermo-Fisher, cat. L3000008), according to the manufacturer's protocol.

### DNA damage to activate p53

To stabilize p53 and induce a p53-mediated transcriptional response, DIvA cells were exposed to DNA damage either by 10-Gy γ -irradiation (using a $^{137}$Cs source, Biobeam 2000) or through the nuclease enzyme AsiSI-RE, activated upon treatment with 300 nM 4-OHT (Sigma-Aldrich, cat. H7904).

### Western blotting

Cells cultured on 10-cm dishes were washed one time in cold PBS and lysed in 300 µL RIPA buffer (25 mM Tris HCl pH 7.6, 150 mM NaCl, 1% Sodium deoxycholate, 1% Triton X-100, 2 mM EDTA dihydrate) supplemented with protease inhibitors (Sigma-Aldrich, cat. 4693124001). Samples were next incubated at 4 °C for 20 min under constant rotation, centrifuged at 12,000 g and the supernatants were then collected. Protein lysates were quantified by BCA assay (Thermo-Fisher, cat. 23225), loaded on 8% or 12% SDS-polyacrylamide gels, and run at 100 V for ~2–3 hours. Proteins were transferred to nitrocellulose membranes in cold transfer buffer (25 mM Tris, 192 mM glycine, 20% methanol) via run at 100 Volts for 2 h at 4 °C. Next, membranes were blocked in 5% non-fat dried milk in TBS-T solution (0.1% Tween20 in TBS: 20 mM Tris base, 137 mM sodium chloride, pH 7.6) for 1 h at RT while shaking. Membranes were incubated with primary antibodies, all diluted in 5% non-fat dried milk in TBS-T solution. The antibodies employed in this study were: mouse monoclonal anti-p53 DO-1 (Santa Cruz Biotechnology, cat. sc-126; 1:3000 dilution, incubated 1 h at RT), rabbit monoclonal anti-p21 (Abcam, cat. ab109520; 1:1000 dilution, incubated overnight at 4 °C), rabbit monoclonal anti-GAPDH (Abcam, cat. ab128915; 1:50,000 dilution, incubated 1 h at RT), rabbit monoclonal anti-NF-κB p65 (Cell Signaling cat. D14E12 XP®; dilution 1:1000), rabbit polyclonal anti-Histone H2B (Abcam cat. ab1790, dilution 1:5000), mouse monoclonal anti-HaloTag (Promega G921A, dilution 1:1000), mouse monoclonal anti-vinculin (Thermo-Fisher, cat. MA5-11690; 1:4000 dilution, incubated 1 h at RT). After antibody hybridization, membranes were washed three times in TBS-T (5 min each wash at RT, while shaking) and incubated for 1 h at RT with peroxidase-conjugated secondary antibodies (anti-mouse IgG, Cell Signaling, cat. 7076; anti-rabbit IgG, Cell Signaling, cat. 7074) diluted 1:5000 in 5% non-fat dried milk in TBS-T solution. The membranes were finally developed using an ECL substrate (Bio Rad, cat. 1705061) and images were acquired with a CCD camera using ChemiDoc MP imaging system.

### RNA extraction and RT-qPCR

Cells grown on 6-cm dishes were washed once in cold PBS and lysed in 750 µL of TRIzol reagent (Thermo-Fisher, cat. 15596018) to extract the total RNA. Lysates were purified using silica membrane columns (Machery-Nagel, NucleoSpin RNA Plus). The isolated RNA was quantified and tested for purity by Epoch Microplate Spectrophotometer (Agilent). For each sample, 2 µg of RNA was reverse-transcribed to cDNA using the High-Capacity cDNA Reverse Transcription Kit (Thermo-Fisher cat. 4368814), following the manufacturer's protocol. Real-time qPCR analysis was performed to assess the expression of p53 target genes. Each reaction (20 µL final volume) was composed as follows: cDNA (5 µL of 1:100 dilution), 150 nM primers, SYBR Green mix

(Roche, LightCycler 480 I Master). To normalize the cDNA amount among different conditions, samples were run altogether using the constitutive gene *GAPDH* as an internal standard.

## Assembly and characterization of the SMT/mSIM microscope

To acquire SMT movies while collecting reference images of the nuclear architecture, we custom-built a microscope capable of both single-molecule imaging of NFs and super-resolved mSIM of a reference channel. The scheme of our SMT/mSIM microscope is depicted in Supplementary Fig. 1a. Briefly, the microscope is composed by two illumination arms, connected to a commercial microscope Frame (Olympus IX-71, Olympus Life Sciences). The first illumination arm, directs the laser light from both a 200 mW 561 nm laser (Cobolt 06-DPL, Hubner Photonics) and a 200 mW 647 nm laser (Coherent Obis, Coherent Inc) to perform SMT using HILO illumination. Here, a movable mirror (MM) in a conjugated plane of the back focal aperture of the objective allows for achieving the desired light beam inclination in the object plane (roughly 67°). The second line expands the collimated light from a 405 nm and a 488 nm lasers (Coherent Obis, Coherent Inc.) to 0.5 cm in size and directs it onto a DMD (Vialux v7000) which creates a pattern of diffraction-limited spots on the sample. The physical size of each DMD micromirror is 13.67 $\mu m$ and, with the lenses used in our microscope (see scheme in Supplementary Fig. 1a), this corresponds to a projected image of each pixel on the sample plane of ~117 nm. The chosen illumination pattern (an equilateral triangular lattice with side equal to 16 DMD pixels) is scanned over the mSIM field of view. 224 different images are necessary to completely scan the entire field of view. The two illumination arms are next combined through a dichroic mirror (DM)(Di03-R488-t3-25 × 36, Semrock Inc.) that directs the excitation light to the sample through a quad-band dichroic (Di03-R405/488/561/635-t3-25 × 36, Semrock inc.) and a 60 × 1.49NA oil-immersion objective (Olympus ApoN 60 × 1.49 Oil, Olympus Inc.). The fluorescent light originating from the sample is collected by the same objective, filtered by the quad-band dichroic and a quad-band emission filter (FF01-466/523/600/677-25, Semrock Inc.) and directed to a sCMOS camera (Orca Fusion C14440-20UP, Hamamatsu Photonics) resulting in an image pixel size equal to 108.3 nm. The microscope is equipped with control systems for temperature (37 °C), $CO_2$ (5%), and humidity, to maintain cells under physiological conditions during live-cell experiments.

We verified that upon reconstruction of the mSIM super-resolved image (see below), the lateral resolution of the microscope increased by a factor higher than 1.4× (Supplementary Fig. 1b), and that minimal chromatic aberration (below the microscope resolution limit, Supplementary Fig. 1c) was present when imaging subdiffraction fluorescent beads with the two different illumination arms of the microscope. Reconstruction of the mSIM image was performed by custom-written routines in Matlab (available at https://github.com/shiner80/Recon_mSIM), as described in Supplementary Note 1.

## SMT/mSIM acquisition

Cells were incubated for 30 min at 37 °C with 1 nM JF$_{549}$ ligand[58] diluted in phenol-red free DMEM. After three washes in PBS, cells were incubated for 10 min at 37 °C with Hoechst 33342 (Thermo-Fisher, cat. H3570, 2 μg/mL in PBS) diluted in phenol-red free DMEM. Cells were washed three times in PBS and fresh phenol-red free DMEM was added. Two mSIM images were collected before and after each SMT movie, projecting on the sample a series of 224 patterns of diffraction-limited spots (one pattern per frame) through the 405 nm laser (49 ms/frame laser exposure, 50 ms/frame camera exposure, 20 fps). The SMT movies were acquired using stroboscopic illumination by the 561 nm laser (5 ms/frame laser exposure, 10 ms/frame camera exposure, 100 fps), collecting 2000 frames per movie. Comparison of the two mSIM acquisitions for each cell allows us to discard those acquisitions affected by cellular motion or stage drift.

## paSMT acquisition

Two days before paSMT experiments, cells were seeded onto 4-well LabTek chambers (Thermo-Fisher, cat. 155382PK) at a ~30% confluence. One hour before imaging, cells expressing HaloTag-conjugated proteins were labeled with two fluorescent ligands, a PA JF$_{549}$[32] for the SMT videos and the readily photoactive JF$_{646}$[58] used for collecting a reference image of the nucleus by a red light-led source (Excelitas Xcite XLED1, Qioptiq). To this end, cells were incubated for 15 min at 37 °C with PA-JF$_{549}$ ligand (10 nM for overexpression experiments, 100 nM for endogenously expressed p53 or CTCF) diluted in phenol-red free DMEM. Next, 10 nM JF$_{646}$ ligand was added to the medium, incubating the cells for a further 15 min at 37 °C. After labeling, cells were thoroughly washed in PBS (two rounds of three washes separated by incubation at 37 °C for 15 min in phenol-red free DMEM).

Photo-activation of the PA-JF$_{549}$ dye was achieved through a 405 nm laser (Coherent, continuous-wave current), with the number of photoactivated molecules (2-3 molecules per frame) tuned by the laser power (0.5-10 mW at the microscope entrance). Photoactivated molecules were excited by a 561 nm laser (Cobolt 60-DPL, 20 mW power at the microscope entrance), adopting stroboscopic illumination to reduce the photobleaching rate and to follow individual molecules for prolonged times. The resulting time-lapse movies (10,000 frames per video) were acquired at 5 ms/frame of laser exposure (561 nm laser) and 10 ms/frame of camera exposure, to obtain 100 frames per second (fps) movies.

## smFISH−sample preparation and image acquisition

Single-molecule Fluorescent In Situ Hybridization (smFISH) was performed to measure nascent and mature RNAs of two p53 target genes (*CDKN1A* and *MDM2*) and the house-keeping gene *GAPDH*. To compare the transcription dynamics of WTp53-HaloTag to p53 mutants, RNA expression was coupled with p53 abundancy quantified via labeling HaloTag during the smFISH protocol. To detect single RNAs, we used HuluFISH RNA probes (PixelBiotech GmbH) -labeled with ATTO-647N fluorophore- for *CDKN1A* and *GAPDH* whereas for *MDM2* we employed Design Ready Stellaris probes (Biosearch Technologies) labeled with Quasar 670 dye. The list of smFISH oligos is provided in Supplementary Table 1. DIvA p53 KO cells were grown on glass coverslips in a 6-well plate and transfected with p53-HaloTag constructs as described above. The day after transfection, cells were fixed in 4% PFA (10 min at RT), and washed with 135 mM Glycine in PBS for 10 min a RT. Next, coverslips were washed twice in PBS and permeabilized with Triton X-100 diluted in PBS at either 0.1% (cells to be hybridized with *MDM2* probes) or 0.3% (*CDKN1A* and *GAPDH*). After two PBS washes, cells were incubated with a fluorescent HaloTag ligand (2.5 μM TMR−Promega cat. G8251- in PBS) for 1 h at RT. To remove the unconjugated TMR, cells were washed with abundant PBS.

### Probe hybridization

**HuluFISH probes (CDKN1A and GAPDH).** Cells were washed twice with HuluWash buffer (2× SSC, 2 M Urea), 10 min at RT. Next, cells were hybridized by adding onto coverslips 0.5 μL of probes (either *CDKN1A* or *GAPDH*) diluted in 50 μL of HuluHyb solution (2× SSC, 2 M Urea, 10% dextran sulfate 5× Denhardt's solution). Hybridization was performed in a humidified chamber at 30 degrees overnight. The following day, cells were washed twice in HuluWash buffer (30 min per wash, at 37 °C in the dark) and then incubated with a DNA staining solution (Hoechst 33342, 1 μg/mL in PBS) for 10 min at RT in the dark. Coverslips were mounted on glass slides with Vectashield (Vector Laboratories) and sealed.

**Design-ready Stellaris probes (MDM2).** Cells were incubated with 1 mL of the washing buffer A, composed as following: 10% saline-sodium citrate (SSC), 20% formamide solution (Thermo-Fisher, ca. AM9342), in RNase-free water (Sigma-Aldrich, ca. 95284). Next, cells

were hybridized by adding 1 µL of 12 nM *MDM2* probe diluted in 100 µL hybridization buffer (10% dextran sulfate, 10% SSC-20× buffer, 20% formamide in RNase-free water). Hybridization was performed by incubating cells in a humidified chamber at 37 °C overnight. Cells were next washed twice in buffer A (30 min per wash, at 37 °C in the dark), once in 10% SCC-20×, and finally incubated with a DNA staining solution (Hoechst 33342, 1 µg/mL in PBS) for 10 min at RT in the dark. Coverslips were mounted on glass slides with Vectashield and sealed.

## smFISH acquisition

Images were acquired with our custom-built widefield microscope (see above), using a ×60 oil-immersion objective (N.A. = 1.49), a sCMOS camera, and a led source for illumination. Z-stacks composed by 34 images were collected with a 0.3 µm step size along the optical axis.

## DNA FISH—sample preparation and image acquisition

DIvA cells seeded in a 24-well plate and grown on coverslip glasses, were washed twice in PBS and fixed using 4% PFA for 10 min at RT. Coverslips were next rinsed three times in PBS at RT (3 min for each wash) and permeabilized using 0.5% Triton X-100 (diluted in PBS) for 10 min at RT. To remove RNAs that may potentially interfere with the procedure, coverslips were treated with 100 µg/mL RNase A (Thermo-Fisher, cat. EN0531) diluted in PBS, for 1 h at RT. Coverslips were then incubated for 1 h with 20% glycerol (in PBS) at RT, followed by three consecutive rounds of freezing, thawing and soaking: 30 sec keeping coverslips on dry ice, gradual thawing in ambient air, and 2-min incubation of 20% glycerol at RT, respectively. Cells were thus washed three times in PBS (10 min each, at RT), and incubated in 0.1 M HCl for 5 min at RT. After washing with 2× SSSC buffer (Sigma-Aldrich, cat. S6639), coverslips were incubated overnight at RT in a solution of 50% formamide (pH 7.0; Thermo-Fischer, cat. AM9342) diluted in 2× SSC. The next day cells were washed in PBS for 3 min, and treated for 2 min at RT with pepsin (40 units/mL; Sigma-Aldrich, cat. P6887) diluted in 10 mM HCl, to break down cellular proteins and facilitate the entering of the probes. Pepsin was next inactivated by two washes with 50 mM Mg2Cl2 in PBS. Cells were then treated with 1% PFA diluted in PBS for 1 min, followed by a washing step in PBS (5 min), and two washes (5 min each) in 2× SSC at RT. Finally, cells were incubated with 50% formamide in 2× SSC, for 1 h at RT.

We used a DNA probe for the *CDKN1A* locus (CHR6: 36644236-36655116) obtained from Genomic Empire (cat. CDKN1A-20-OR), and labeled with 5-TAMRA (orange dUTP, 548 nm). A probe mix solution was prepared by adding 2 µL of *CDKN1A* probe and 8 µL of hybridization buffer (Genomic Empire). The mix was denatured at 73 °C for 5 min, followed by 2 min in ice, and next left for 15 min at 37 °C. In parallel, coverslips with cells were warmed at 73 °C for 5 minutes, to denature genomic DNA. After denaturation, 10 µL/sample of the probe mixture was added on a pre-heated slide (73 °C). A coverslip with cells was placed on top of the drop, with cells facing the probe. Coverslips were sealed with rubber cement probes, to avoid the hybridization mix from drying out. Cells were then placed into a humidified chamber, and incubated at 37 °C for 16 hours (in the dark) to allow hybridization. The next day, rubber cement was removed from slides and the coverslips were washed with solution A, composed as following indicated: 0.3% Igepal (Sigma, cat. CA-630) diluted in 0.4× SSC. Cells were incubated with solution A at 73 °C for 2 min (in the dark), while gently shaking. A second wash was performed with the solution B (0.1% Igepal in 2× SSC) for 2 min (in the dark) at RT, while shaking. Cells were next incubated with a DNA staining solution (Hoechst 33342, 1 µg/mL in PBS) for 10 min at RT in the dark. After three washes in PBS, coverslips were mounted on slides using Vectashield and sealed with nail polish.

Images of DNA FISH were collected using our custom-built microscope (see the previous section), with a ×60 oil-immersion objective (N.A. = 1.49) and a sCMOS camera. Chromatinized nuclei (stained by Hoechst) were acquired with our high-resolution

implementation (mSIM), using a 405 nm laser (49 ms/frame laser exposure, 50 ms/frame camera exposure, 20 fps), collecting 224 images per cell, as described above.

## FRAP

Fluorescence recovery after photobleaching (FRAP) experiments were carried out on a Leica SP-8 confocal microscope equipped with a ×63 NA 1.4 oil-immersion objective, an Argon laser and a white-light laser (WLL). The pixel size was set to 160.1 nm, and timelapses were collected on regions of 256 × 256 pixels, with a frame time equal to 262 ms. Pre-bleach and post-bleach imaging were collected setting the WLL wavelength at 552 nm, a laser power of 70% and an AOTF value of 7%. The emitted fluorescent signal was collected with a PMT in the spectral bandwidth of 565 nm to 700 nm. Bleaching was performed in a circular region of 2 µm in diameter by increasing the WLL AOTF to 100% and by additionally delivering 70% of the 488 nm laser line of the Argon laser. 10 pre-bleach images, 2 bleach iterations and 500 post-bleach images were collected. FRAP data was analyzed by using available MATLAB routines[59].

## Quantification and statistical analysis
### paSMT analysis

**Tracking.** SMT movies from paSMT acquisitions were analyzed using the ImageJ plugin TrackMate that detects individual spots in each frame of a movie and connects them into tracks[60]. The diffraction-limited spots corresponding to single TF molecules are identified by applying a LoG (Laplacian of Gaussian) filter to each frame to suppress noise and signal arising from large structures (e.g., cellular auto-fluorescence, out of focus background). Spot detection on the LoG filtered images was performed in TrackMate by setting 0.8 µm as maximum molecule diameter and an intensity threshold of 5. The identified spots are then connected into tracks, using the LAP algorithm[61] that requires to specify the maximal frame to frame displacement for the observed molecular specie. The probability of molecule to diffuse more than a certain distance $r_{max}$ in a time $\Delta t$ can be calculated as:

$$P(r > r_{max}, \Delta t) = e^{\frac{-r_{max}^2}{4D\Delta t}} \quad (1)$$

Where $D$ is the diffusion coefficient of the observed molecule. The maximal frame-to-frame displacement in TrackMate was therefore set using the relationship above, so that less than 1% of the tracks would be missed for typical diffusion coefficients of nuclear proteins (smaller than $5\mu m^2/s$)[62]. This leads to a maximal displacement threshold $r_{max} = 1\mu m$ in a frame-to-frame time $\Delta t = 10ms$. The control experiments on unconjugated HaloTag were instead analyzed with a maximal displacement $r_{max} = 2\mu m$, since its diffusion coefficient has been reported in the range of $15\mu m^2/s$[62].

TrackMate generated tables providing lists with x, y positions and the time for each track. The tables were imported in MATLAB and the tracks were analyzed using custom-written routines to: (i) generate and model the distribution of single-molecule displacements—in order to extract the abundance of each kinetic subpopulation (indicated as bound, slow and fast fractions in the result section), and their respective diffusion coefficients; (ii) classify individual track segments in one of the three subpopulations using an Hidden Markov model approach, vbSPT; (iii) analyze the co-clustering between bound molecules and diffusing molecules; (iv) analyze the diffusional aniso-tropy of unbound molecules.

**Distribution of single-molecule displacements.** The tracks generated by TrackMate were used to populate a histogram of frame-to-frame displacements, using a bin size $\Delta r$ equal to 20 nm. The histogram was then normalized (to a probability density function, PDF) to give the probability $p(r)\Delta r$ to observe a molecule moving a distance

between $r - \frac{\Delta r}{2}$ and $r + \frac{\Delta r}{2}$ in the time $\Delta t$. This probability is then fit by a three-component diffusion model:

$$p(r)\Delta r = r\Delta r \sum_{i=1}^{3} \frac{f_i}{2D_i\Delta t} \exp\left(\frac{-r^2}{4D_i\Delta t}\right) p(\Delta z, D_i) \qquad (2)$$

Where $f_i$ is the fraction of molecules with diffusion coefficient $D_i$, so that $\sum_{i=1}^{3} f_i = 1$.

In this formulation $D_i$ assumes the value of apparent diffusion coefficient given by the sum of the true diffusion coefficient and a term due to the limited localization accuracy $\sigma^2$ in SMT: $D_i = D_{i,true} + \frac{\sigma^2}{\Delta t}$.

The term $p(\Delta z, D_i)$ accounts for the probability that a molecule has to remain within the excitation slice, with thickness $2\Delta z$ which can be calculated as (see Supplementary Note 2):

$$p(\Delta z, D) = \frac{1}{2\Delta z}\left(2\Delta z\, \text{erf}\left(\frac{\Delta z}{\sqrt{D\Delta t}}\right) + \sqrt{\frac{4D\Delta t}{\pi}}e^{-\frac{(\Delta z)^2}{D\Delta t}} - \sqrt{\frac{4D\Delta t}{\pi}}\right) \qquad (3)$$

Fitting of the cumulative distribution of displacements was performed at the single cell level, and the distributions of obtained parameters ($D_{bound}, D_{slow}, D_{fast}, f_{bound}, f_{slow}$) were plotted as violin-plots using the IoSR-Surrey MATLAB toolbox, and compared using Kruskal–Wallis non-parametric testing. p-values were Bonferroni adjusted for multiple comparisons on the 5 different parameters ($D_{bound}, D_{slow}, D_{fast}, f_{bound}, f_{slow}$).

**Track classification.** To classify p53 tracks into 'Bound', 'Slow diffusing' and 'Fast diffusing' segments, we accumulated data on different cells, isolated tracks lasting longer than 8 frames and gave them as input to the vbSPT algorithm[34], by imposing a maximum number of states equal to 3. The algorithm provides the diffusion coefficients for each of the states together with the transition rates between these states. We verified that the diffusion coefficients extracted for each isolated component are in agreement with the ones obtained by the analysis of displacements. Further, we verified that no contamination of bound molecules is present in the diffusing components extracted by vbSPT, which is fundamental in order to correctly analyze diffusional anisotropy.

**Co-clustering between bound and diffusing molecules.** We evaluated the co-clustering between bound and free molecules by first calculating the centroid of each of the track segments classified as bound and then calculating the cross-correlation between these centroids and all the particle positions belonging to 'slow diffusing' and 'fast diffusing' track segments. These histograms were then compared to the distribution of distances between bound molecules and randomly positioned molecules within the cell nucleus.

**Analysis of diffusional anisotropy.** To analyze diffusional anisotropy, we removed bound track segments using vbSPT, and we calculated the angle between two consecutive displacements. We then calculated the fold-anisotropy metric $f_{180/0}$, a measure of the likelihood that a single molecule will be in the backward direction, compared to taking a step forward–calculated as the probability of observing a backward jump (with an angle between jumps in the range $[180° - 30°, 180° + 30°]$), divided by the probability of observing a forward jump (with an angle between jumps in the range $[0° - 30°, 0° + 30°]$). Plotting $f_{180/0}$ as function of the distance run by the molecule allows to discriminate between compact, non-compact and guided exploration[19]. Error bars were calculated as standard deviation from 100 subsampling of the data using 50% of the original data.

## SMT/mSIM analysis
Analysis of the combined SMT/mSIM acquisitions were carried out with custom-written MATLAB routines. First, in order to correct for the illumination pattern–dimmer at the FOV edge than at its center–to provide a map for local variations in chromatin density,–the reconstructed mSIM images are normalized, by dividing the mSIM image by a blurred version (with a gaussian filter with a standard deviation of 2.16 μm) of it. Next, TF tracks are generated from the SMT movie using TrackMate as described above. In analogy with the definition of Brownian diffusion coefficients in 2D, $D = \frac{<r^2>}{4Dt}$ –where $<r^2>$ is the average squared displacement–we calculated an 'instantaneous' diffusion coefficient $D_{inst}$, for each recorded single-molecule displacement $r$ as $D_{inst} = r^2/4D\Delta t$, and we related these $D_{inst}$ to the DNA density class, i.e., the quartile of normalized Hoechst intensity found at the starting position of each displacement. Heat-maps such as those presented in Fig. 1g are then calculated by computing the frequency with which displacements with a given $D_{inst}$ fall in each of the four DNA density classes.

For HaloTag-p53 we also analyzed the relative DNA density surrounding p53 molecules belonging to different diffusion states. To this end, single-molecule tracks were sent to the vbSPT algorithm to extract track segments belonging to the 'bound', 'slow diffusion' and 'fast diffusion' states. These classified track segments are then overlaid to the mSIM image of DNA density. Next, the average normalized Hoechst intensity in a $1\,\mu m \times 1\,\mu m$ pixel region surrounding each molecule is calculated separately for each of the three diffusion states of the TF.

## smFISH analysis
smFISH data was analyzed using the MATLAB package FISH-quant. Mature RNA molecules are identified as 3D Gaussian spots with maximal intensity above an arbitrary threshold, that was kept constant for each of the target RNAs analyzed. Nascent RNAs at active transcription sites were identified by looking for high intensity nuclear foci setting the detection threshold so that no more than four actively transcribed loci could be found in each nucleus. This analysis allows to count the number of active transcription sites per cell and to count the number of nascent RNAs present at the transcription site at the moment of fixation[63].

## Materials availability
Cell lines and plasmids generated for this study are available from the corresponding author with a completed materials transfer agreement.

## Reporting summary
Further information on research design is available in the Nature Portfolio Reporting Summary linked to this article.

# Data availability
The data to reproduce all plots of this study are provided in the Source Data file. Due to the large data volume, raw microscopy data will be provided upon request. Source data are provided with this paper.

# Code availability
Original Matlab code to reconstruct mSIM images has been deposited at https://github.com/shiner80/Recon_mSIM and is publicly available as of the date of publication[64]. Any additional information required to reanalyze the data reported in this paper is available from the corresponding author upon request.

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

## Acknowledgements

We thank professors A.S. Hansen and S. Zambrano for their insightful comments on the manuscript. Acquisition of FRAP data was carried out in ALEMBIC, Ospedale San Raffaele. The research leading to these results has received funding from AIRC under the IG 2018-ID:21897 project—P.I. Davide Mazza and from Worldwide Cancer Research, Grant Reference number 22-0116—P.I. Davide Mazza. T.F. is funded by a Maria Sklodowska-Curie Innovative Training Network (ITN-PEP-NET, Grant agreement ID: 813282). Part of the present work was performed by M.M., E.C., and T.F. in partial fulfillment of the requirements for obtaining the PhD degree at Università Vita-Salute San Raffaele, Milano, Italy.

## Author contributions

M.M., A.L., D.G., and P.F. performed experimental studies. M.M., A.L., S.C., E.B, G.L. generated reagents and biological models. E.C., Z.L., and D.M. designed, assembled and validated the mSIM microscope. E.C and D.M. wrote the software to reconstruct/analyze the mSIM data. M.M., A.L., E.C., T.F., E.M., and D.M. carried out the analysis. C.T. and D.M. supervised the work. D.M. designed the project and acquired funds. M.M., A.L., and D.M. wrote the manuscript. All authors reviewed and edited the manuscript.

## Competing interests

The authors declare no competing interest
