## [Peer Review File · Nature Communications]

Chromatin organization drives the search mechanism of nuclear factorsREVIEWER COMMENTS

Reviewer #1 (Remarks to the Author):

Mazzocca and Loffreda et al set to address the important question of how nuclear factors scan the genome and, particularly, what is the role of chromatin organization in directing this search. This is a timely topic; the data is very interesting and would be useful for researchers in chromatin biophysics, transcriptional regulation and nuclear organisation. My review mainly addresses aspects of chromatin biophysics, nuclear organisation and transcriptional regulation. I hope that the technical aspects of imaging will be covered by other reviewers with more specific expertise in that.

A strength of this manuscript is the comparison between the diffusion of different nuclear factors within the context of chromatin density and, later in the manuscript, the link to transcriptional regulation. A weak point of this manuscript is the tendency of the authors to extend their conclusions beyond what is directly supported by the data. By doing so, the authors commonly undermined or ignore alternative explanations for some of their observations. For instance, I am not convinced that all the observations in this manuscript are directly attributed to a "search" mechanism rather than to chromatin dynamics. I am also not convinced that a size exclusion model can be excluded as a determinant of search mechanism(s), as some of the data in the manuscript actually fit a size exclusion model. But I believe that the data here is important, so the manuscript should be suitable for publication if the reviewers will be able to moderate unsupported statements or better substantiate them according to the points below.

Major points

1. Chromatin dynamics as a potential mechanism for determining the localisation of nuclear factors: In Fig 1d-g, the unconjugated HaloTag was found to be enriched in regions of lower DNA density and H2B. But is this really directly related to the chromatin density? Could the authors exclude the possibility that the dynamics of chromatin are a driving factor for determining the baseline occupancy of macromolecules rather than DNA/chromatin density? To clarify, by "chromatin dynamics" I am referring to aspects such as the movement and diffusing of nucleosome clutches within TADs, histone exchange, rapid movement of RNA and macromolecules at DNA low-density regions etc. Could the authors use the H2A data to quantify the local diffusion of H2A as a proxy for chromatin dynamics? If yes, then they should repeat the correlative analyses in Fig 1d-g using that measure of chromatin dynamics. Otherwise, the authors should minimally discuss chromatin dynamics as a potential mechanism that could affect the nuclear distribution of factors or specify why they believe this might not be relevant.

2. Possible link between the molecular weight (MW) of nuclear factors and their accessibility to chromatin: Correlative analysis in Fig 1e led the authors to conclude that the "differences in protein accessibility to DNA dense nuclear regions did not appear to be caused by size exclusion effects, as no correlation between protein molecular weight and enrichment in higher DNA classes was observed (Fig. 1e)". But is this really true? If the authors would exclude the single outlier from the analysis in Fig 1e (second point from the left, with MW = ~50 kDa), wouldn't they obtain a positive correlation? When MW was estimated, did the authors bring into account the oligomerisation state that might have previously been reported for some of these factors? On the same rationale, wouldn't all the observations in Fig 6 be explained by FUS-p53 molecular assemblies ("condensates") being too large to penetrate heterochromatic regions and therefore cannot activate p53 target genes? It is highly recommended to address these questions, at least textually, as in my view the data cannot exclude a size exclusion effect, and to some extent actually support it.

3. The interpretation of the anisotropy analysis as a "search" (Fig 2 and 3c): The anisotropy analysis was summarised by the authors accordingly: "In summary, factors displaying different enrichment in

DNA dense regions display also different search strategies". This statement was made after the authors observed high anisotropy for the factors that enriched at dense DNA regions (CTCF and H2B). Yet, can the authors exclude the possibility that these observations are rather attributed to chromatin dynamics? Specifically, is it possible that CTCF and H2B seem to move back and forth (i.e. anisotropy) simply because the chromatin they bound to move that way? Given that histone exchange in heterochromatin is rather low, how can the apparent movement of H2B even be explained as a "search", mechanistically? The authors should either exclude the possibility that anisotropy is attributed to chromatin dynamics, or otherwise they should moderate related statements textually.

4. Ectopic expression of factors: All the experiments in Fig 1 and 2 were carried out using ectopically expressed factors. But how is the expression level of these ectopically expressed factors compared with the expression level of the endogenous factors? If the ectopic expression level is much higher than the endogenous level, then there is a concern that some of the observations in Fig 1 and 2 are driven by oligomerization and/or non-specific targeting. The authors should address that.

5. Poor modelling of the fast diffusion coefficients in Fig 3: Three diffusion coefficients were modelled and quantified in Fig 3a and then used across this figure. But frankly, I don't see trimodal distribution in either of the histograms in Fig 3a. Instead, what I see there is one distinct slow diffusion coefficient and then a broad distribution of fast diffusion coefficients. The authors should EITHER collect better data that would allow them to demonstrate trimodal distribution OR find a more convincing way to model the distribution coefficient without assuming three diffusion coefficients. In the latter case, they will need to adjust the modes and analyses in Fig 3d and Fig 4a-b because these are made based on three diffusion coefficients, which is a model that is not supported by the data.

Minor points

6. The word "strategy" is commonly used across the manuscript, including in the title ("exploration strategy"). But "strategy" is an ambiguous word when attributed to nuclear factors, as it has no obvious biophysical meaning. For that reason, the usage of the word "strategy" is confusing, especially when placed in the title. For clarity, it is best to stick to terms that are more commonly used in biophysical studies.

7. A citation is needed on page 2, to substantiate the statement: "a timescale that seems incompatible with the classical diffusion-limited search mechanism."

8. In Fig 1e-f, it would be helpful to mark each dot in these scatter plots by the name of the protein it represents.

Reviewer #2 (Remarks to the Author):

Pls see attached PDF

In this paper, Mazzocca et al. combined multifocal SIM with SMT to investigate how nuclear factors use different search strategies to explore chromatin and find their genomic targets. By correlating live-cell NF dynamics with super-resolution chromatin density mapping and characterizing parameters such as instantaneous diffusion coefficient and diffusional anisotropy, they showed that NFs with different DNA-binding propensities display distinct accessibility to DNA-dense regions. Specifically, they showed that the search mechanism of p53 alternates between a fast, non-compact exploration mode and a slow, exhaustive sampling mode, modulated by DNA density as well as its interaction with chromatin via its DNA-binding domain and terminal IDRs. They also found that this guided exploration mechanism can be effectively hampered by removing the C-terminal IDR and specific DNA-interacting residues, whereas introducing IDR from the FUS protein can restore and even enhance this mechanism. The disrupted genome targeting associated with IDR deletions was further correlated to a lowered ability to induce transcription, as evidenced by the expression of two p53 targets. Finally, the authors showed that the enhanced search associated with FUS-p53 can be abrogated upon IDR-mediated condensate formation of the protein, thereby demonstrating the important role played by IDR-mediated interactions in both determining as well as fine-tuning the search strategy employed by NFs.

Overall, the experimental approaches adopted in this work are sound and well-designed, the measurements and analyses are solidly performed, and the conclusions are convincingly substantiated by the data. I did not spot any major issue regarding logic, data or argumentation. I do have, however, a variety of issues concerning the presentation (e.g. lack of proper description/explanation or clear labelling, inconsistent usage of specific terms, and imprecise expressions that could potentially lead to misinterpretations), all of which tend to undermine the clarity and persuasiveness of the manuscript, and for which revisions are required.

Specific points:

- The term NF- κ B is used in the text while p65 is used in the figures. Please stick to 1 term.
- The description of the diffusion coefficient as 'speed' is not advisable as they are two distinct entities.
- P. 8: "...is described by three apparent diffusion coefficients, with medians equal to 0.07, 0.9 and 4.4 $\mu\text{m}^2/\text{s}$ respectively (Fig. 3b)." Please indicate if these numbers refer to measurements before or after p53 activation by DNA damage, add the S.D. to each value, and perhaps label them as $D_{\text{bound}}/D_{\text{slow}}/D_{\text{fast}}$ (in line with the labeling used in Fig. 3b).
- P. 11: "...interactions (mediated by the p53 IDRs and by its binding domain) control the capability of p53 to...." Please specify the type(s) of interactions involved.
- P. 12: Please specify what is meant by "more markedly". What is the critical concentration?
- P. 25: "...thresholding value of 5." It is not clear what the threshold refers to.
- P. 27: The description of the instantaneous diffusion coefficient is not clear. Please give a more detailed explanation why the squared displacement is divided by $4\Delta t$ and add an equation. Also, "dividing for" should read "dividing by".

Figure-related issues:

- Some of the plots are missing definition of the error bar in the corresponding legend.
- Pls add a scale bar to the images in Figs. 1b, 2a and 6b.

- In all figures with plots of diffusion coefficients, all columns have the same width. However, due to the use of log scale, the columns on the right represent 10x more data points than those on the left. This can be rectified by plotting instead $\log(D/\mu\text{m}^2\text{s}^{-1})$.
- The violin plots in Figs. 3b and 6d are missing label on the vertical axis. Also, it is strange to put the unit (e.g. $\mu\text{m}^2/\text{s}$) in the plot title on top. Lastly, the statistical features represented in the violin plots should be specified in the legend.
- Fig. 1c: Please specify whether the images shown for p53 refer to the p53 (NT) or p53 (IR) state.
- Fig. 1e, f: Please indicate which symbol refers to which NF. Also, the vertical axis label should read "Enrichment in DNA dense regions".
- Fig. 2a: The '+' sign on the left-hand side of the equation is not being followed by any chemical species. Also please explain 'max proj' and the dotted line in blue in the legend.
- Fig. 2b: The use of the term "fold" in the equation is incorrect: the ratio in the bracket is the "fold"!
- Fig. 3a: Please indicate the statistical distribution used to fit the histograms.
- Fig. 4b: It seems that changes in Hoechst intensity is only $\leq 5\%$; is that truly significant? Please justify. Also, please mark the positions of the p53 molecules in the images, as the images only show chromatin density.
- Fig. 4d: Why are there error bars for the Hoechst plot but not the DNA FISH plot? Also, it is not clear how is the normalization done in each plot, and which parameters are being tested by the t-test.
- Fig. 5a: It might be good to indicate the positions of the residues mutated in the p53-mSB mutant.

Expression-related issues:

- There are a lot of spelling inconsistencies throughout the manuscript (e.g. scale bar vs. scale-bar or scale bar-; 4-OHT vs. 4OHT; (μm) vs. [μ m] or [μm], etc.; N/C-terminus vs. N/C terminus; NT vs. untreated, etc). Pls thoroughly check and rectify.
- P. 8: "...since a p53 mutant at the seven residues responsible for specific DNA contacts (p53-mSB) displays a significant reduction in the fraction of molecules involved in such component..." should read "...since a p53 mutant with mutations at the seven residues responsible for specific DNA contacts (p53-mSB) displays a significant reduction in the fraction of molecules involved in that component..."
- P. 10: "...displayed reduced backward diffusional anisotropy than p53 WT..." should read "...displayed a reduced backward diffusional anisotropy compared to p53 WT..."
- P. 14: "...can speed up the search mechanism..." is not correct: you can speed up the search or the rate, but not a mechanism!
- P. 18: "Plasmids were transiently transfected in p53-null cells" should read "p53-null cells were transiently transfected with plasmids."
- P. 25: "The probability of a diffusing molecule to displace..." should read "The probability of a molecule to diffuse...".
 "lists with x, y and t positions": Does *t* refer to a position or to time?
 "...probability density function, pdf..." should read "...probability density function, PDF...", in line with the axis label in the figures.

Reviewer #3 (Remarks to the Author):

Mazzocca and Loffreda et al. have developed a custom-built microscope that enables single molecule tracking of transcription factors, while also imaging nuclear density at super-resolution using structured illumination microscopy. Their study focused on investigating the search process of the transcription factor p53, which must locate target sites within the vast genome in addition to activate genes. Through careful quantification, the researchers discovered that p53 utilizes two distinct modes of diffusion depending on the density of DNA. In regions with low DNA density, p53 exhibits a fast, non-compact exploration mode, whereas in DNA-dense regions, it adopts a slower, guided exploration mode to expedite the search process. The study also revealed the significance of intrinsically disordered regions (IDRs) in p53 for guided exploration. Mutants of p53 with less IDRs demonstrated reduced propensity for guided exploration, leading to less expression of target genes. Conversely, a p53 mutant with additional IDRs exhibited an increased propensity for guided exploration, resulting in upregulated expression of target genes at low expression levels. Notably, when p53 with additional IDRs was expressed at high levels, it led to the formation of condensates, preventing its localization in DNA-dense regions and subsequently less target gene expression.

The research was conducted meticulously with appropriate controls, and the paper is well-written and clear, reflecting the high standards expected from the Mazza group. The investigation into the role of condensates in gene expression regulation is a currently trending topic, and this work significantly contributes to the field.

Minor comments:

The authors may consider discussing the result that an $f_{180/0}$ value of ~ 0.9 for p65 implies directed motion.

Although the IDRs of p53-mSB are intact, the mutant still demonstrates impaired guided exploration. The authors should discuss how specific binding residues could contribute to the guided exploration mode.

It would be interesting to compare the fluorescent intensities at transcription sites between the wild-type (WT) and deltaC p53. This analysis could provide insights into whether non-specific binding affects transcriptional activity.

The authors should include violin plots of the fast fraction, particularly in Figure 6d, as the manuscript refers to this data.

The citation 46 has been published in eLife and should be updated accordingly.

Dear Editor and Reviewers,

We would like to thank you for the thorough review of our manuscript and for the positive reception. We have tried to address all the points raised by the referees, by performing additional experiments and/or analyses and by editing the manuscript – as detailed in the point-by-point reply to reviewer below. Our replies are highlighted in blue. Changes in the manuscript text are highlighted in yellow.

Reviewer #1 (Remarks to the Author):

Mazzocca and Loffreda et al set to address the important question of how nuclear factors scan the genome and, particularly, what is the role of chromatin organization in directing this search. This is a timely topic; the data is very interesting and would be useful for researchers in chromatin biophysics, transcriptional regulation and nuclear organisation. My review mainly addresses aspects of chromatin biophysics, nuclear organisation and transcriptional regulation. I hope that the technical aspects of imaging will be covered by other reviewers with more specific expertise in that.

A strength of this manuscript is the comparison between the diffusion of different nuclear factors within the context of chromatin density and, later in the manuscript, the link to transcriptional regulation. A weak point of this manuscript is the tendency of the authors to extend their conclusions beyond what is directly supported by the data. By doing so, the authors commonly undermined or ignore alternative explanations for some of their observations. For instance, I am not convinced that all the observations in this manuscript are directly attributed to a “search” mechanism rather than to chromatin dynamics. I am also not convinced that a size exclusion model can be excluded as a determinant of search mechanism(s), as some of the data in the manuscript actually fit a size exclusion model. But I believe that the data here is important, so the manuscript should be suitable for publication if the reviewers will be able to moderate unsupported statements or better substantiate them according to the points below.

We are grateful to the referee for carefully reviewing our manuscript, and we are happy to hear that they judge our results relevant. We have tried to address the referee comments extensively, by performing additional experiments and analyses, and by editing the results and discussion sections of the manuscript to reflect what is supported by our data more closely, as detailed below point-by-point.

Major points

1. Chromatin dynamics as a potential mechanism for determining the localisation of nuclear factors: In Fig 1d-g, the unconjugated HaloTag was found to be enriched in regions of lower DNA density and H2B. But is this really directly related to the chromatin density? Could the authors exclude the possibility that the dynamics of chromatin are a driving factor for determining the baseline occupancy of macromolecules rather than DNA/chromatin density? To clarify, by “chromatin dynamics” I am referring to aspects such as the movement and diffusing of nucleosome clutches within TADs, histone exchange, rapid movement of RNA and macromolecules at DNA low-density regions etc. Could the authors use the H2A data to quantify the local diffusion of H2A as a proxy for chromatin dynamics? If yes, then they should repeat the correlative analyses in Fig 1d-g using that measure of chromatin dynamics. Otherwise, the authors should minimally discuss chromatin dynamics as a potential mechanism that could affect the nuclear distribution of factors or specify why they believe this might not be relevant.

Indeed, several papers support the hypothesis that chromatin mobility might affect the accessibility of soluble nuclear factors. For example, Hihara et al. (Cell Reports, 2012) has shown that hindering chromatin mobility by cross-linking could impair accessibility to dense chromatin regions. Direct measurement of the role of chromatin mobility in factor accessibility would require to follow both chromatin dynamics and single

molecule nuclear factor mobility simultaneously, at high temporal resolution. Unfortunately, our SMT/mSIM microscope does not currently allow for high-speed two-color time lapses.

To verify whether the DNA density provided by our mSIM images correlates with chromatin mobility, we re-analyzed our histone H2B SMT/mSIM data – as suggested by the reviewer - measuring whether histones incorporated in regions at higher DNA density display different mobility than those in regions at lower DNA density. To this end we classified H2B tracks depending on the class of chromatin density in which they resided, filtered out those H2B molecules that moved across regions of different DNA density and only kept those tracks that lasted longer than 25 frames (to exclude freely diffusing H2B molecules that diffuse out of focus in few frames). We analyzed the mean squared displacement plot of these tracks, which - as expected - was well described by an anomalous diffusion model, $MSD = 4D_{\alpha}t^{\alpha}$. All MSD curves were well fitted by this model, providing anomalous exponents $0.35 < \alpha < 0.5$ in good agreement with data on chromatin mobility (e.g. Oliveira GM et al, Nat Comm, 2021). However, when comparing the mobility of histones incorporated in different classes of DNA density (Class 1 = lower DNA density, class 4 = higher DNA density) no clear trend was evident, potentially in agreement with previous reports that highlighted that the dynamics of chromatin fluctuations do not correlate with chromatin density (Audugè et al, Nucl Ac Res, 2019).

Thus, while it is still possible that chromatin dynamics will influence the steady-state occupation of chromatin dense regions, our data cannot provide direct evidence to this hypothesis. Nevertheless, we now discuss this possibility in the manuscript (p 16), and we hope that future implementations of our microscope, to allow simultaneous imaging of fast time-lapses of chromatin dynamics at high resolution and diffusivity measurements of soluble factors, will enable to dissect the relationship between chromatin movement and nuclear factor accessibility.

2. Possible link between the molecular weight (MW) of nuclear factors and their accessibility to chromatin: Correlative analysis in Fig 1e led the authors to conclude that the “differences in protein accessibility to DNA dense nuclear regions did not appear to be caused by size exclusion effects, as no correlation between protein molecular weight and enrichment in higher DNA classes was observed (Fig. 1e)”. But is this really true? If the authors would exclude the single outlier from the analysis in Fig 1e (second point from the left,

with MW = ~50 kDa), wouldn't they obtain a positive correlation? When MW was estimated, did the authors bring into account the oligomerisation state that might have previously been reported for some of these factors? On the same rationale, wouldn't all the observations in Fig 6 be explained by FUS-p53 molecular assemblies ("condensates") being too large to penetrate heterochromatic regions and therefore cannot activate p53 target genes? It is highly recommended to address these questions, at least textually, as in my view the data cannot exclude a size exclusion effect, and to some extent actually support it.

We agree that our original sentence (at page 5 of the current manuscript version) was misleading. Our data indeed do not show that the size/molecular weight has no role in determining the factors accessibility, but rather that differences in protein size are not sufficient to explain the different accessibility of different proteins. We have reworded that sentence, and we discussed the role of size-exclusion more in detail in the discussion. Additionally we have performed the additional controls proposed by the referee, in particular:

- a) We have re-analyzed the correlation accounting for the oligomerization state of different proteins. The results now shown in Supplementary Figure 2b indicate that even when accounting for protein oligomerization no clear correlation between protein size and accessibility to chromatin dense domains emerge.
- b) We have repeated the analysis removing the 'outlier' (histone H2B). As shown on the left, after removing the histone H2B, a slight positive correlation is observed, but such correlation is not statistically significant. We prefer not to include such plot in the manuscript since we cannot find a justifiable reason for excluding H2B from this plot.

As shown on the left, after removing the histone H2B, a slight positive correlation is observed, but such correlation is not statistically significant. We prefer not to include such plot in the manuscript since we cannot find a justifiable reason for excluding H2B from this plot.

An additional evidence for the limited role of volume exclusion effects in the accessibility of different NFs to chromatin derives from current work in our lab. We are trying to develop a minimal biophysical/mathematical model to reproduce the measured chromatin occupancy in DNA dense regions for different factors. Following the approach developed in Isaacson S.A. et al, PNAS, 2011, we started modeling the effect of chromatin density (estimated by structured illumination microscopy) on the diffusion of nuclear factors as a pure volume exclusion effect, and we

found that this model is unable to reproduce the chromatin occupancy plots observed in our work. Instead, accounting for the interactions between the factors and chromatin significantly improves the model predictions. Such model, that we are still refining, will be the object of a follow-up manuscript, focusing on the theoretical/modelling aspects of diffusion in chromatin.

3. The interpretation of the anisotropy analysis as a "search" (Fig 2 and 3c): The anisotropy analysis was summarised by the authors accordingly: "In summary, factors displaying different enrichment in DNA dense regions display also different search strategies". This statement was made after the authors observed high anisotropy for the factors that enriched at dense DNA regions (CTCF and H2B). Yet, can the authors exclude the possibility that these observations are rather attributed to chromatin dynamics? Specifically, is it possible that CTCF and H2B seem to move back and forth (i.e. anisotropy) simply because the chromatin they bound to move that way? Given that histone exchange in heterochromatin is rather low, how can the apparent movement of H2B even be explained as a "search", mechanistically? The authors should either exclude the possibility that anisotropy is attributed to chromatin dynamics, or otherwise they should moderate related statements textually.

We agree with the referee: chromatin bound molecules would naturally result in a high anisotropy, because of two reasons (a) Chromatin motion is confined; (b) The precision of localization affects the measurement of anisotropy on short displacements. These confounding effects and the strategies to minimize the biases that they might cause are described in detail in the paper by Hansen et al. (Nat Chem Biol, 2020).

In our paper we followed the guidelines of the Hansen paper: in order to minimize the impact of chromatin bound molecules on the measurements of diffusional anisotropy, we filtered out those molecules being classified as “bound” by the vbSPT algorithm - before performing the anisotropy analysis. Supplementary Figure 2c, displays the distribution of displacements of the nuclear factors before and after filtering out these bound molecules. Of note, after such filtering, CTCF and histone H2B molecules, residual bound fractions for H2B and CTCF are $<10^{-10}$. Additionally, after such filtering the distributions of displacements for H2B and CTCF are both described by two components with diffusion coefficients on the order of $0.7\mu\text{m}^2/\text{s}$ and $\sim 5\mu\text{m}^2/\text{s}$ respectively, both significantly higher than the diffusion coefficients reported for chromatin loci and chromatin bound proteins ($<0.1\mu\text{m}^2/\text{s}$) (see for example: Chen et. al Cell, 2014, Morisaki et al., Nat Comm, 2014).

We thus believe that we can reasonably exclude that the observed anisotropy is caused by binding to chromatin: in agreement with this statement, our results on CTCF diffusional anisotropy, completely overlap with the analysis performed by Hansen et al., 2020: in that case mutant analysis allowed to determine that CTCF anisotropy was not caused by binding to chromatin but to transient trapping in ~ 100 nm size regions mediated by interactions with RNA. We now discuss this point at page 14 of the manuscript.

4. Ectopic expression of factors: All the experiments in Fig 1 and 2 were carried out using ectopically expressed factors. But how is the expression level of these ectopically expressed factors compared with the expression level of the endogenous factors? If the ectopic expression level is much higher than the endogenous level, then there is a concern that some of the observations in Fig 1 and 2 are driven by oligomerization and/or non-specific targeting. The authors should address that.

We are grateful to the reviewer for highlighting this important point. We agree that overexpression of fluorescently tagged nuclear factors could in principle perturb their dynamics and their localization in chromatin. Of the factors that we tested, unconjugated HaloTag is by definition overexpressed as no endogenous counterpart exist. For CTCF-HaloTag we have used a previously validated U2OS knock-in cell line (Hansen et al, 2020), so the tagged protein is expressed at endogenous levels. For p53-HaloTag we report comparable single molecule results both in overexpression conditions (Figure 1-2) and for knock-in cell lines (Figure 3 and Supplementary Figure 3). For the two remaining factors, H2B-HaloTag and p65-HaloTag, we measured the fold expression level of the exogenous tagged protein on the endogenous ones by western blot. HaloTag-H2B is expressed at much lower levels than the endogenous H2B, to a degree that is difficult to quantify since the endogenous histone H2B is so abundant that its signal is already heavily saturated when the HaloTag-H2B band appears. HaloTag-p65 is instead moderately (1.4 fold) overexpressed with respect to the endogenous p65. In previous papers focusing on live-cell microscopy, HaloTag-p65 was similarly overexpressed with no impact on its function or its macroscopic dynamics (Callegari et al, Plos Genetics, 2019; Zambrano et al. elife, 2020). We now add these western blots to Supplementary Figure 2a.

5. Poor modelling of the fast diffusion coefficients in Fig 3: Three diffusion coefficients were modelled and quantified in Fig 3a and then used across this figure. But frankly, I don't see trimodal distribution in either of the histograms in Fig 3a. Instead, what I see there is one distinct slow diffusion coefficient and then a broad distribution of fast diffusion coefficients. The authors should EITHER collect better data that would allow them to demonstrate trimodal distribution OR find a more convincing way to model the distribution coefficient without assuming three diffusion coefficients. In the latter case, they will need to adjust the

modes and analyses in Fig 3d and Fig 4a-b because these are made based on three diffusion coefficients, which is a model that is not supported by the data.

Indeed, in Fig 3a, we plot the *distributions of the displacements* (not the *distribution of diffusion coefficients*) jumped by p53 molecules between consecutive frames and we report that this distribution can be described by three Brownian diffusion populations representing bound, slow and fast diffusing molecules respectively.

We note that we do not expect to clearly identify three ‘modes’ in such distributions of displacements, since the distribution that describes these displacements for Brownian diffusion (as given by Eq. 2 in the methods section), is long-tailed. Modelling the distribution of displacements is a standard approach in intranuclear single molecule tracking since it is more robust than classical methods (such as MSD analysis) when dealing for short tracks and since it allows to correct for the biases caused by having an observation slice of limited thickness (see for example Mazza et al., Nucl Ac Res. 2012, Hansen et al., eLife, 2017).

The observation that p53 distribution of displacements is described by three components is in agreement with our previous single molecule tracking data on p53 (Mazza et al., 2012, Morisaki et al, 2014, Loffreda et al., 2017). In those papers we have shown that p53 diffusion in the nucleus of living cells is not described satisfactorily by just two components (one bound + one freely diffusing), but we need at least three (one bound + two diffusing).

Following the referee comments, we now add (Supplementary Figure 3e) a quantitative analysis of the minimal model required to fit p53 mobility. We extracted diffusion coefficients from individual tracks by MSD analysis, we fit the experimental distribution of diffusion coefficients with a Gaussian mixture with an increasing number of components: model selection based on the Bayesian information criterion (BIC) confirms that a model with three components (one bound plus two diffusing) is the most suited to describe the experimental data, resulting in a minimum BIC.

Yet, we do agree with the referee that the ‘slow-diffusing’ and the ‘fast-diffusing’ populations most likely represent ‘effective’ populations, and cannot be strictly interpreted as two distinct ‘species’ of p53 molecules. Likely the most correct representation of p53 motion is, as the referee mentions, a ‘spectrum’ of diffusion coefficients. Approaches to extract ‘diffusion coefficient spectra’ from SPT data exist (Heckert et al, eLife, 2022), but these methods do not allow to classify track segments into subpopulation since they do not assign an individual diffusion coefficient to individual tracks segments.

Classification of p53 molecules into “bound”, “slow-diffusing” and “fast-diffusing” groups, allowed us instead to analyze separately segments of the single-molecule tracks (thanks to the vbSPT algorithm) and to recognize that molecules moving slower have indeed higher propensity to access DNA dense regions. We have edited the text order to better explain the rationale behind classification of p53 molecules into three dynamic classes, both in the results section (page 8) and in the discussion section (page 14).

Minor points

6. The word “strategy” is commonly used across the manuscript, including in the title (“exploration strategy”). But “strategy” is an ambiguous word when attributed to nuclear factors, as it has no obvious biophysical meaning. For that reason, the usage of the word “strategy” is confusing, especially when placed in the title. For clarity, it is best to stick to terms that are more commonly used in biophysical studies.

We have modified the term ‘exploration strategy’ into ‘search mechanism’ which has a more recognizable meaning in the biophysics field.

7. A citation is needed on page 2, to substantiate the statement: “a timescale that seems incompatible with the classical diffusion-limited search mechanism”.

Thanks for spotting this, we now added an appropriate citation.

8. In Fig 1e-f, it would be helpful to mark each dot in these scatter plots by the name of the protein it represents.

We have modified the panels to display the proteins represented by each data point.

Reviewer # 2

In this paper, Mazzocca et al. combined multifocal SIM with SMT to investigate how nuclear factors use different search strategies to explore chromatin and find their genomic targets. By correlating live-cell NF dynamics with super-resolution chromatin density mapping and characterizing parameters such as instantaneous diffusion coefficient and diffusional anisotropy, they showed that NFs with different DNA-binding propensities display distinct accessibility to DNA-dense regions. Specifically, they showed that the search mechanism of p53 alternates between a fast, non-compact exploration mode and a slow, exhaustive sampling mode, modulated by DNA density as well as its interaction with chromatin via its DNA-binding domain and terminal IDRs. They also found that this guided exploration mechanism can be effectively hampered by removing the C-terminal IDR and specific DNA-interacting residues, whereas introducing IDR from the FUS protein can restore and even enhance this mechanism. The disrupted genome targeting associated with IDR deletions was further correlated to a lowered ability to induce transcription, as evidenced by the expression of two p53 targets. Finally, the authors showed that the enhanced search associated with FUS-p53 can be abrogated upon IDR-mediated condensate formation of the protein, thereby demonstrating the important role played by IDR-mediated interactions in both determining as well as fine-tuning the search strategy employed by NFs. Overall, the experimental approaches adopted in this work are sound and well-designed, the measurements and analyses are solidly performed, and the conclusions are convincingly substantiated by the data. I did not spot any major issue regarding logic, data or argumentation. I do have, however, a variety of issues concerning the presentation (e.g. lack of proper description/explanation or clear labelling, inconsistent usage of specific terms, and imprecise expressions that could potentially lead to misinterpretations), all of which tend to undermine the clarity and persuasiveness of the manuscript, and for which revisions are required.

We thank the reviewer for the positive reception of our manuscript and for providing important insights on how to improve it. We have performed some additional analysis of our data and modified the text and the figures of the manuscript to address the reviewer comments, as detailed in the point by point response below.

Specific points:

- The term NF- κ B is used in the text while p65 is used in the figures. Please stick to 1 term.

We now consistently refer to p65 throughout the manuscript and the figures.

- The description of the diffusion coefficient as 'speed' is not advisable as they are two distinct entities.

We have modified the text of the manuscript referring to diffusion coefficients instead of speed.

- P. 8: "...is described by three apparent diffusion coefficients, with medians equal to 0.07, 0.9 and 4.4 $\mu\text{m}^2/\text{s}$ respectively (Fig. 3b)." Please indicate if these numbers refer to measurements before or after p53 activation by DNA damage, add the S.D. to each value, and perhaps label them as $D_{\text{bound}}/D_{\text{slow}}/D_{\text{fast}}$ (in line with the labeling used in Fig. 3b).

We have added the median and the standard deviation for the diffusion coefficients of p53 both for before and after activation by DNA damage.

- P. 11: "...interactions (mediated by the p53 IDRs and by its binding domain) control the capability of p53 to..." Please specify the type(s) of interactions involved.

We can only speculate that the interactions involved are non-specific interactions with chromatin, given that we see a correlation between the factors bound fraction and the occupation of DNA dense regions. We therefore decided to rephrase the sentence into "Thus, in agreement with our previous results on different NFs, those p53 mutants displaying lower bound fraction (such as IDR lacking mutants) are also less capable of being targeted to regions at higher DNA density." that in our opinion reflects more fairly the correct interpretation of our data.

- P. 12: Please specify what is meant by "more markedly". What is the critical concentration?

The concept that we wanted to deliver in this section of the results is that the transcriptional response for increasing nuclear levels of p53-WT and FUS-p53 is different. Transcription of p53 target genes progressively increase with increasing p53-WT levels. For FUS-p53 we instead observe a biphasic behavior: p53-FUS induces more transcripts than WT-p53 (up to 3-4 times more, depending on the gene) up to a certain nuclear level, but above this 'threshold', further increasing the levels of FUS-p53 result in reduced transcription of the target genes. We have now reworded this section to try to improve the clarity. We also agree with the referee that the phrasing 'critical concentration' can be misleading, since our measurement of HaloTag-p53 fluorescence intensity is only proportional to the p53 concentration. We changed this expression into 'nuclear level threshold', that better reflects the nature of our data.

- P. 25: "...thresholding value of 5." It is not clear what the threshold refers to.

The threshold is an intensity threshold applied to the DoG filtered images. We now explain this more in detail in the methods section.

- P. 27: The description of the instantaneous diffusion coefficient is not clear. Please give a more detailed explanation why the squared displacement is divided by $4\Delta t$ and add an equation. Also, "dividing for" should read "dividing by".

For Brownian diffusion process in 2D the average squared displacement $\langle r^2 \rangle$ is related to the diffusion coefficient D by the relationship $\langle r^2 \rangle = 4D\Delta t$, where Δt is the time interval over which the displacement takes place. Inversion of this relationship provides $D = \langle r^2 \rangle / 4\Delta t$. To provide estimates of local mobility, we thus apply the same relationship to individual displacements, leading to $D_{\text{inst}} = r^2 / 4\Delta t$. We now clarify this in the methods section of the paper (page 29).

Figure-related issues:

- Some of the plots are missing definition of the error bar in the corresponding legend.

We now added the definition of the error bars and sample size in all the legends.

- Pls add a scale bar to the images in Figs. 1b, 2a and 6b.

We added the missing scale bars.

- In all figures with plots of diffusion coefficients, all columns have the same width. However, due to the use of log scale, the columns on the right represent 10x more data points than those on the left. This can be rectified by plotting instead $\log(D/\mu\text{m}^2\text{s}^{-1})$.

Thanks for pointing this out. We re-plot the instantaneous diffusion vs chromatin class plots as function of $\log(D/\mu\text{m}^2\text{s}^{-1})$. Indeed, this is how already these plots were calculated (in log scale), so only evident difference is the label of the x-axis in these plots.

- The violin plots in Figs. 3b and 6d are missing label on the vertical axis. Also, it is strange to put the unit (e.g. $\mu\text{m}^2/\text{s}$) in the plot title on top. Lastly, the statistical features represented in the violin plots should be specified in the legend.

We have added y-axes labels to these plots, and removed the units of measurement from the plot titles. We now include the statistical features of the violin plots in the corresponding figure legends.

- Fig. 1c: Please specify whether the images shown for p53 refer to the p53 (NT) or p53 (IR) state.

The image refers to p53(IR). We now added the label to the figure.

- Fig. 1e, f: Please indicate which symbol refers to which NF. Also, the vertical axis label should read "Enrichment in DNA dense regions".

We have added labels to each individual point in Fig 1e, f. We have modified the vertical axis label into "Enrichment in DNA dense Regions".

- Fig. 2a: The '+' sign on the left-hand side of the equation is not being followed by any chemical species. Also please explain 'max proj' and the dotted line in blue in the legend.

We corrected the scheme of the chemical reaction representing HaloTag labelling and photoactivation by 405nm light in Fig 2a. We now explain in the figure legend what "max proj" and the dotted line represent.

- Fig. 2b: The use of the term "fold" in the equation is incorrect: the ratio in the bracket is the "fold"!

We have changed the equation in fig2b and we modified the figure legend, to indicate more clearly how the f180/0 ratio is calculated, i.e. as the ratio between the probability of observing backward jumps over forward jumps.

- Fig. 3a: Please indicate the statistical distribution used to fit the histograms.

The distribution used to fit the data is reported in Eq. 2 of the methods. We now refer to this equation in the legend of Fig 3a.

- Fig. 4b: It seems that changes in Hoechst intensity is only $\leq 5\%$; is that truly significant? Please justify. Also, please mark the positions of the p53 molecules in the images, as the images only show chromatin density.

We thank the referee for pointing this out. The plot in Fig 4b, represents the average chromatin density surrounding individual p53 molecules, an average that is performed over tens of thousands of p53 molecules. The plot thus shows that –on average – p53 molecules with different mobility are preferentially positioned in regions with different chromatin density. However, we observe – for example – many bound p53 molecules that sit at edges of chromatin dense regions (so that chromatin will be denser on one side of the molecule and less dense on the other one'. When averaging these molecules together, the net effect is to bring the differences between the different populations down). Yet this difference is statistically significant (we now add a statistical test in Fig 4b) and reproducible across repetition of the experiment. We also added a mark indicating the position of the p53 molecules in the figure.

- Fig. 4d: Why are there error bars for the Hoechst plot but not the DNA FISH plot?

Error bars were present on both the Hoechst and on the DNA FISH plot. However since these plots are calculated by averaging together the signals around the DNA FISH plot (distance = 0, correspond to the center of the FISH spot), it is expected that the variability of the DNA FISH plot will be very small, since we are basically averaging together many almost-identical signals. To allow an appreciation of the variability of the FISH signal, we now plot the symbols of the plot as 'empty circles', that allow to visualize the (small) error bars also for the FISH data.

Also, it is not clear how is the normalization done in each plot, and which parameters are being tested by the t-test.

The plots in Fig 4b and Fig 4d are all normalized in the same way, as fold change compared to the average Hoechst or DNA FISH signal in the nucleus. The one-sample t-test is calculated on the point at shorter distance of the plot (i.e. at the position of p53 molecules/DNA FISH spot), to verify that this point is found at a chromatin density that is significantly different from 1 (i.e. from the average Hoechst intensity in the nucleus). We now better clarify these points in the figure legend.

- Fig. 5a: It might be good to indicate the positions of the residues mutated in the p53-mSB mutant.

We added the mutated residues to the scheme of the p53-mSB mutant.

Expression-related issues:

- There are a lot of spelling inconsistencies throughout the manuscript (e.g. scale bar vs. scale-bar or scale bar-; 4-OHT vs. 4OHT; (μm) vs. [$\mu\text{ m}$] or [μm], etc.; N/C-terminus vs. N/C terminus; NT vs. untreated, etc). Pls thoroughly check and rectify.

We have attempted to thoroughly correct spelling inconsistencies in the text and in the figures of the manuscript.

- P. 8: "...since a p53 mutant at the seven residues responsible for specific DNA contacts (p53-mSB) displays a significant reduction in the fraction of molecules involved in such component..." should read "...since a p53 mutant with mutations at the seven residues responsible for specific DNA contacts (p53-mSB) displays a significant reduction in the fraction of molecules involved in that component..."

Thanks, we revised the sentence according to your suggestion.

- P. 10: "...displayed reduced backward diffusional anisotropy than p53 WT..." should read "...displayed a reduced backward diffusional anisotropy compared to p53 WT..."

Thanks, we revised the sentence according to your suggestion.

- P. 14: "...can speed up the search mechanism..." is not correct: you can speed up the search or the rate, but not a mechanism!

Thanks, we revised the sentence according to your suggestion.

- P. 18: "Plasmids were transiently transfected in p53-null cells" should read "p53-null cells

Thanks, we revised the sentence according to your suggestion.

transiently transfected with plasmids."

Thanks, we revised the sentence according to your suggestion.

- P. 25: "The probability of a diffusing molecule to displace..." should read "The probability of a molecule to diffuse..."

Thanks, we revised the sentence according to your suggestion.

"lists with x, y and t positions": Does t refer to a position or to time?

Yes, we have reworded the sentence to make it clearer.

"...probability density function, pdf..." should read "...probability density function, PDF...", in line with the axis label in the figures.

Thanks, we revised the sentence according to your suggestion.

Reviewer #3 (Remarks to the Author):

Mazzocca and Loffreda et al. have developed a custom-built microscope that enables single molecule tracking of transcription factors, while also imaging nuclear density at super-resolution using structured illumination microscopy. Their study focused on investigating the search process of the transcription factor p53, which must locate target sites within the vast genome in addition to activate genes. Through careful quantification, the researchers discovered that p53 utilizes two distinct modes of diffusion depending on the density of DNA. In regions with low DNA density, p53 exhibits a fast, non-compact exploration mode, whereas in DNA-dense regions, it adopts a slower, guided exploration mode to expedite the search process. The study also revealed the significance of intrinsically disordered regions (IDRs) in p53 for guided exploration. Mutants of p53 with less IDRs demonstrated reduced propensity for guided exploration, leading to less expression of target genes. Conversely, a p53 mutant with additional IDRs exhibited an increased propensity for guided exploration, resulting in upregulated expression of target genes at low expression levels. Notably, when p53 with additional IDRs was expressed at high levels, it led to the formation of condensates, preventing its localization in DNA-dense regions and subsequently less target gene expression.

The research was conducted meticulously with appropriate controls, and the paper is well-written and clear, reflecting the high standards expected from the Mazza group. The investigation into the role of condensates in gene expression regulation is a currently trending topic, and this work significantly contributes to the field.

We are grateful to the reviewer for their positive evaluation of our manuscript and for their important suggestions, that we have tried to address as discussed in the point-by-point response below.

Minor comments:

The authors may consider discussing the result that an $f_{180/0}$ value of ~ 0.9 for p65 implies directed motion.

We now comment that positive $f_{180/0}$ values for p65 might reflect directed motion, although we can only speculate on the mechanisms causing this behavior. One possibility is that non-specific interactions in the interchromatin domain can funnel p65 diffusion. A second possibility might be related to recent evidence about the directed motion of molecular motors such as myosin in the nucleus of eukaryotic cells. (Hari-Gupta et al, Nat Comm 2022).

Although the IDRs of p53-mSB are intact, the mutant still demonstrates impaired guided exploration. The authors should discuss how specific binding residues could contribute to the guided exploration mode.

We now explicitly discuss that the DNA binding domain is also involved in the guided exploration process (page 15). This is probably not surprising since in-vitro studies have demonstrated that the DNA binding domain of p53 takes an active part in the facilitated diffusion process on isolated DNA, together with the C-terminal IDR.

It would be interesting to compare the fluorescent intensities at transcription sites between the wild-type (WT) and deltaC p53. This analysis could provide insights into whether non-specific binding affects transcriptional activity.

We thank the referee for suggesting this additional analysis. Indeed, the brightness of the transcription sites (TS) reports for the number of nascent RNA being produced at these sites at the time of cell fixation. Higher number of nascent transcripts per TS could be in principle be a consequence of (a) higher amplitude of transcriptional bursts and/or (b) higher frequency of transcriptional initiation (if the transcription initiation rate is faster than the release rate of mRNAs from the TS). Performing this analysis on CDKN1A and MDM2 transcripts provides different outcomes. For CDKN1A, the number of nascent RNA per active TS does not depend on the nuclear levels of p53. Further p53 WT and p53-deltaC appear to stimulate the same amount of nascent transcription. We interpret this data in this way: assuming that for CDKN1A the transcription initiation rate is slower than the mRNA release rate, both versions of p53 are capable of stimulating transcription at the same level, once they reach the CDKN1A locus, but p53 WT reaches this locus more frequently (as evidenced by higher number of active TS/cells) than p53-deltaC.

For the MDM2 gene the results are more complex. In this case, higher p53 levels result in a higher number of nascent MDM2 RNAs per TS. This could be caused by either a cooperative action of multiple p53 oligomers at the MDM2 promoter, or by assuming that the MDM2 mRNA release rate from the TS is slower than the transcription initiation rate. Interestingly nascent MDM2 RNAs increase more sharply with increasing levels of p53 WT than p53-d30. Thus also for MDM2 the results on nascent mRNA can be reconciled with the hypothesis that p53 WT can reach its binding sites more frequently than p53-delta30. We now have added these results to the manuscript (Figure 5f).

The authors should include violin plots of the fast fraction, particularly in Figure 6d, as the manuscript refers to this data.

We have added the violin plots of the fast fraction in Figure 3b and in Figure 6d.

The citation 46 has been published in eLife and should be updated accordingly.

Thanks for spotting this. We have now updated the reference to report its published status.

REVIEWERS' COMMENTS

Reviewer #1 (Remarks to the Author):

The authors addressed all the points that I brought up and I have no further major concerns.

Reviewer #2 (Remarks to the Author):

The authors have satisfactorily addressed my concerns. I recommend publication of this work.

Reviewer #3 (Remarks to the Author):

The authors have addressed all of my comments. One minor thing is in the new Fig. 5f, they may want to add in the MDM2 label for clarity.

Dear Editor, we are happy to hear that the reviewers were satisfied by our revisions, and we thank them once more for their valuable feedback. Please find below our point-by-point reply (in green).

REVIEWERS' COMMENTS

Reviewer #1 (Remarks to the Author):

The authors addressed all the points that I brought up and I have no further major concerns.

We would like to thank Reviewer #1 for their positive reception of our manuscript.

Reviewer #2 (Remarks to the Author):

The authors have satisfactorily addressed my concerns. I recommend publication of this work.

We would like to thank Reviewer #2 for their positive reception of our manuscript.

Reviewer #3 (Remarks to the Author):

The authors have addressed all of my comments. One minor thing is in the new Fig. 5f, they may want to add in the MDM2 label for clarity.

We would like to thank Reviewer # 3 for their positive reception of our manuscript. We have added the labels to figure 5F.